# ViPRA: Video Prediction for Robot Actions

**Sandeep Routray**[1,2]    **Hengkai Pan**[1]    **Unnat Jain**[2,3†]    **Shikhar Bahl**[2†]    **Deepak Pathak**[1,2†]

[1]Carnegie Mellon University    [2]Skild AI    [3]University of California, Irvine
[†]Denotes joint mentoring

## Abstract

Can we turn a video prediction model into a robot policy? Videos, including those of humans or teleoperated robots, capture rich physical interactions. However, most of them lack labeled actions, which limits their use in robot learning. We present **Video Prediction for Robot Actions** (ViPRA), a simple pretraining-finetuning framework that learns continuous robot control from these actionless videos. Instead of directly predicting actions, we train a video-language model to predict *both future visual observations and motion-centric latent actions*, which serve as intermediate representations of scene dynamics. We train these latent actions using perceptual losses and optical flow consistency to ensure they reflect physically grounded behavior. For downstream control, we introduce a chunked *flow matching decoder* that maps latent actions to robot-specific continuous action sequences, using only 100 to 200 teleoperated demonstrations. This approach avoids expensive action annotation, supports generalization across embodiments, and enables smooth, high-frequency continuous control upto $22\,\mathrm{Hz}$ via chunked action decoding. Unlike prior latent action works that treat pretraining as autoregressive policy learning, ViPRA explicitly models both what changes and how. Our method outperforms strong baselines, with a 16% gain on the SIMPLER benchmark and a 13% improvement across real world manipulation tasks. We have released models and code at `https://vipra-project.github.io`.

## 1 Introduction

Robots learn by doing, but collecting robot demonstrations, particularly at scale, is expensive, time-consuming, and limited by embodiment. In contrast, videos are abundant. From YouTube clips (Abu-El-Haija et al., 2016) of people performing tasks (Grauman et al., 2022; 2024; Goyal et al., 2017; Damen et al., 2018) to logs of teleoperated robots (Collaboration et al., 2023), they capture rich physical interactions, diverse objects, and long-horizon behaviors that are difficult to script or reproduce. The challenge is that most of these videos may not include action labels.

At the same time, recent advances in video prediction models (Liu et al., 2024; Blattmann et al., 2023; Singer et al., 2022; Zhou et al., 2022; NVIDIA et al., 2025a) open up a new opportunity: learning directly from large corpora of *actionless videos*. Beyond preserving high-level task semantics, these generative models exhibit a strong grasp of object dynamics and fine-grained physical interactions. This naturally leads to a central question: **Can a video prediction model be transformed into a control policy for physical robots?** In this work, we explore this question through a simple and scalable pretraining-finetuning framework that adapts a powerful video-language model (Liu et al., 2024) into a robot policy capable of learning from passive videos.

During pretraining, we co-train on two intuitive objectives: (i) predicting *what* happens next, in the form of *future visual observations*, and (ii) predicting *how* the scene evolves, using a compact intermediate representation known as *latent actions*[1]. By training with both objectives, the model learns to capture both semantic intent and physical dynamics. In contrast, prior latent action pretraining methods (Ye et al., 2024; Bu et al., 2025; Chen et al., 2024; NVIDIA et al., 2025b) treat

---

[1]Latent actions can be viewed as action-like tokens that summarize the transition between states without requiring access to ground-truth control commands

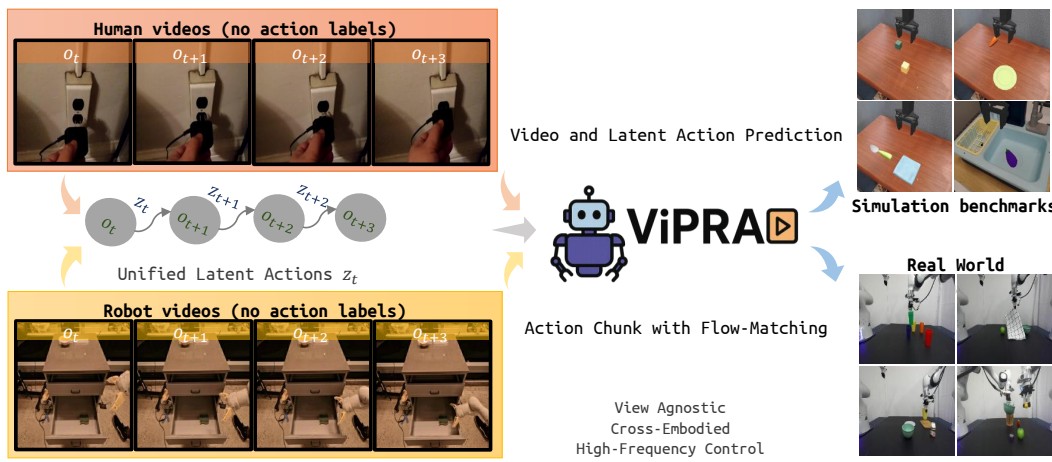

Figure 1: We present ViPRA, a framework to learn generalist robot policies from large-scale actionless videos. It extracts motion-centric latent action sequences, pretrains a video-language model to jointly predict future visual observations and latent action chunks, and finetunes a flow matching decoder to map these latents to smooth, continuous action chunks with minimal labeled data for smooth, high-frequency control.

pretraining purely as policy learning in latent space, without leveraging video prediction or modeling state transitions, and often use temporally coarse task-centric latent actions. Our framework instead predicts state transitions through video prediction and outputs a sequence of fine-grained *motion-centric* latent actions (3 to 6 Hz) over short horizons, capturing high-frequency dynamics critical for control. We further incorporate optical flow consistency as an additional supervision signal, promoting physically plausible and motion-aware latent representations.

Importantly, our pretraining leverages both unlabeled human and robot videos, which enables generalization across embodiments (see Figure 1, left). This broad exposure to passive visual data sets the foundation for effective finetuning with only a small number of teleoperated robot demonstrations. For finetuning on these demonstrations, we employ a *flow matching decoder* (Lipman et al., 2022) that maps latent actions to smooth, continuous robot action chunks (see Figure 1, right). Unlike prior vision-language-action models (VLAs) (Zitkovich et al., 2023; Kim et al., 2024; Li et al., 2024a; Black et al., 2024; Qu et al., 2025), which required thousands of hours of labeled action trajectories[2], our decoder aligns latent transitions with embodiment-specific motor behaviors. This design amortizes inference latency via *action chunking*, enabling smooth, high-frequency control by producing multiple low-level actions in a single forward pass. Our policy can support control rates upto 22 Hz, to our knowledge matched only by one other 7B-parameter model (Kim et al., 2025).

In summary, our contributions are as follows.

(i) A scalable method to extract fine-grained motion-centric latent actions from unlabeled human and robot videos using perceptual and optical flow consistency losses.

(ii) A novel pretraining framework for robot control that jointly predicts future visual states and motion-centric latent actions within a unified video-language model.

(iii) A data-efficient pretraining–finetuning framework that integrates flow matching and action chunking to enable smooth, high-frequency continuous control, operating at up to 22 Hz.

(iv) Demonstrate empirical gains of 16% on the SIMPLER benchmark (Li et al., 2024b) and 13% on real world tasks over the strongest prior continuous control baselines.

## 2 RELATED WORK

**Vision-Language-Action Models** VLAs (Zitkovich et al., 2023; Kim et al., 2024; Li et al., 2024a; Black et al., 2024; Qu et al., 2025) extend vision-language models (VLMs) (Touvron et al., 2023; Chen et al., 2023; Driess et al., 2023; Karamcheti et al., 2024; Beyer et al., 2024) by imitation learning on action-labeled robot demonstrations (Collaboration et al., 2023). Recent works explore auxiliary objectives including visual trace prediction (Niu et al., 2024), chain-of-thought reason-

---

[2]10000 hours of pretraining data (Collaboration et al., 2023) and 5-100 hours of finetuning demonstrations

ing (Wei et al., 2022), and conversational instruction tuning (Li et al., 2025b). However, all existing VLAs require extensive labeled action data, creating a fundamental scalability bottleneck due to the prohibitive cost of data collection. Furthermore, these models focus primarily on grounding language in visual semantics while lacking explicit mechanisms for modeling physical dynamics or temporal structure in action generation. In contrast, ViPRA eliminates the labeled data requirement by leveraging unlabeled videos during pretraining and incorporates temporal dynamics through joint prediction of future visual states and multi-step latent actions, which provides robust priors for high-frequency control.

**Robot Learning from Videos** Videos offer a scalable source of information about object dynamics, task structure, and human behavior. Visual planning methods (Du et al., 2023a; Wu et al., 2024a; Ko et al., 2024; Du et al., 2023b; Baker et al., 2022; Liang et al., 2025b; Luo et al., 2025a) use generative video models to plan in video or video-language space and rely on an inverse dynamics model to convert predicted frames into actions. While effective for long-horizon reasoning, these methods often incur high inference costs, limiting their suitability for high-frequency, dexterous control. Different from the above, policy supervision approaches (Luo et al., 2025a;b) use video models as supervision or reward sources to train policies, though evaluations have so far focused on simulation or relatively constrained real world tasks.

Recent work explores joint training for video generation and action prediction (Li et al., 2025a; Guo et al., 2024), with Li et al. (2025a) also introducing decoupled action decoding to mitigate inference overhead, but evaluations are mostly on smaller-scale datasets, simulation, and do not demonstrate scaling to internet-scale passive videos.

Other efforts leverage human videos to pretrain visual representations for downstream visuomotor control (Nair et al., 2023; Dasari et al., 2023; Xiao et al., 2022; Karamcheti et al., 2023), or extract intermediate cues such as affordances (Bahl et al., 2023; Kannan et al., 2023; Bharadhwaj et al., 2023; Liu et al., 2022; Goyal et al., 2022), interactions (Zeng et al., 2024), or visual traces (Wen et al., 2024; Bharadhwaj et al., 2024; Mandikal & Grauman, 2022; Bahl et al., 2022) from unlabeled videos to guide policy learning. These approaches depend on structured priors or explicit cue extraction, which can constrain scalability. In contrast, we learn motion-centric latent actions that capture temporal dynamics and align them with video-language representations, enabling scalable learning from large action-free video corpora.

**Latent Action Spaces** Latent action representations improve data efficiency by enabling learning from action-free videos via self-supervised learning (Dwibedi et al., 2018; Liang et al., 2025a; Seo et al., 2022; Schmidt & Jiang, 2024; Cui et al., 2024). Recent methods impose discrete information bottlenecks with vector quantization encoders (van den Oord et al., 2017) and predict these tokens during policy learning (Ye et al., 2024; Lee et al., 2024; Yang et al., 2024a; Bu et al., 2025; Chen et al., 2024; NVIDIA et al., 2025b), achieving strong real world results through imitation. Some train inverse dynamics models on limited labeled demonstrations before applying them to unlabeled video (Baker et al., 2022; Xu et al., 2023), while others treat latent actions as abstract embodiments and jointly train policies with inverse dynamics model predictions across embodiments (Jang et al., 2025). Another line of work uses these abstractions to build world simulators (Gao et al., 2025; Bruce et al., 2024) or plan in latent spaces (Ha & Schmidhuber, 2018; Racanière et al., 2017; Hafner et al., 2019a;b; Lee et al., 2020; Wu et al., 2023; Sekar et al., 2020). While these methods capture physical dynamics effectively, they struggle to generalize to novel settings due to limited semantic grounding. Video-language models can provide such multimodal grounding (Du et al., 2023a; Ko et al., 2024; Liang et al., 2025b; Guo et al., 2024; Du et al., 2023b), but existing approaches are typically computationally heavy and slow at inference. In contrast to existing methods, ViPRA learns fine-grained, motion-centric latent actions that capture temporal dynamics while leveraging a video-language model (Liu et al., 2024) for semantic grounding. We train a unified latent space from large-scale, action-free human and robot videos, enabling cross-embodiment transfer. By predicting action chunks during both latent pretraining and real-action finetuning, we amortize inference latency and achieve smooth, high-frequency control.

# 3 BACKGROUND

We defer discussion on VQ-VAE for discrete latent actions, optical flow estimation, behavior cloning, and flow matching for continuous control to Appendix A.

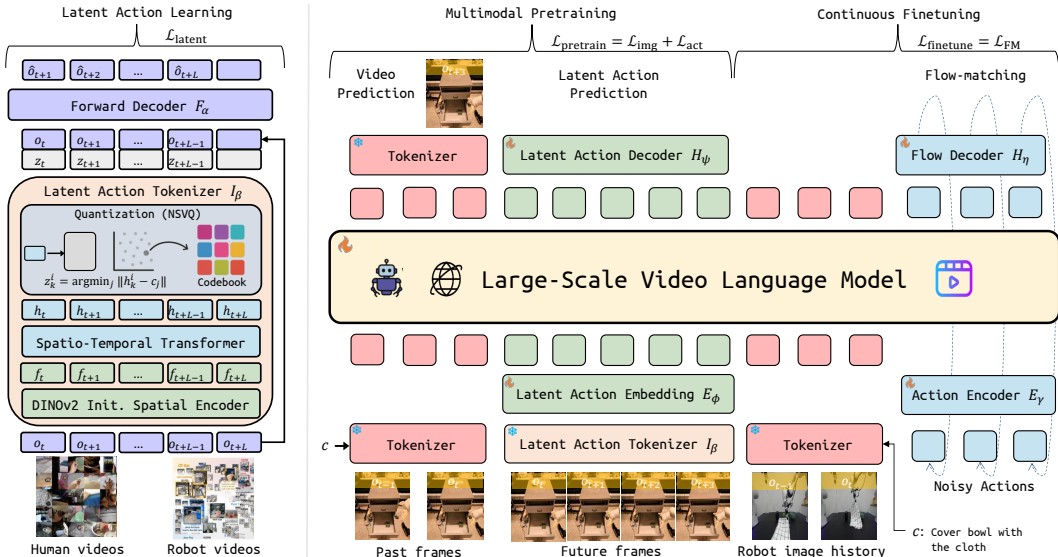

Figure 2: **ViPRA framework** comprises of: (1) **Latent Action Learning** (left): A neural quantization bottleneck extracts discrete latent action sequences $z_{t:t+L-1}$ provided image sequences $o_{t:t+L}$ from actionless human and robot videos, trained via latent loss $\mathcal{L}_{\text{latent}}$ to capture motion-centric dynamics. (2) **Multimodal Pretraining** (center): A video-language model jointly predicts future observations $o_{t+H}$ and latent action sequences $z_{t:t+H-1}$ from past frames $(o_{t-1}, o_t)$ and task description $c$, using loss $\mathcal{L}_{\text{pretrain}}$. (3) **Continuous Finetuning** (right): A flow matching decoder maps latent actions to continuous robot actions $a_{t:t+H-1}$ using noisy action conditioning and loss $\mathcal{L}_{\text{FM}}$, enabling smooth, high-frequency control.

# 4  VIDEO PREDICTION FOR ROBOT ACTIONS

A generalist robotic agent must combine precise low-level control with environment-agnostic high-level intelligence. Video generation models are well-suited to this goal, as future-state prediction captures both physical interaction detail and task-related semantic context. Achieving this requires effectively utilizing large-scale data, architectures that expose motion-centric signals, and stable training pipelines. To this end, we present ViPRA: *(i)* learning motion-centric discrete latent actions from large-scale human and robot videos without action supervision, guided by perceptual and optical flow consistency; *(ii)* pretraining a multimodal video-language model to jointly predict future visual observations and latent action sequences, enabling semantic grounding of temporal dynamics; and *(iii)* finetuning a flow matching decoder that maps latent actions to smooth, continuous control using only a few hundred demonstrations. This hierarchy leverages the rich physical priors of video models while ensuring the precision needed for real world robot control.

## 4.1  LATENT ACTION LEARNING FROM ACTIONLESS VIDEOS

We first train a latent action model to represent behavior in both human and robot videos without requiring action supervision. As illustrated in Fig. 2 (left), this phase extracts motion-centric latent action sequences that capture how the environment changes (inverse dynamics) and reciprocally also models the visual observations in response to these implicit actions (forward prediction).

Given a length-$(L + 1)$ observation sequence $o_{t:t+L}$ sampled from human or robot videos, our objective is to learn latent action sequences $z_{t:t+L-1}$ where each $z_k = [z_k^1, z_k^2, \ldots, z_k^{N_{\text{latent}}}]$ uses $N_{\text{latent}}$ discrete tokens to encode the motion dynamics at timestep $k$. Here, each latent action component $z_k^i$ is quantized from a shared codebook $\mathcal{C}$ of size $|\mathcal{C}| = 8$.

We train an inverse dynamics encoder $I_\beta(z_{t:t+L-1} \mid o_{t:t+L})$ that predicts latent action token $z_t$ by conditioning on the full observation sequence $o_{t:t+L}$. This non-causal design allows each latent action $z_k$ to incorporate both past and future context, making it sensitive to local motion intent–for instance, distinguishing a pickup from a putdown based on surrounding frames. By providing the encoder with the entire clip, we reduce reconstruction ambiguity and force $z_k$ to encode the minimal but sufficient information to explain the local transition.

We jointly train a forward decoder $F_\alpha(\hat{o}_{t+k+1} \mid o_{t:t+k}, z_{t:t+k})$ that predicts the future frame $\hat{o}_{t+k+1}$ given the history of observations $o_{t:t+k}$ and latent actions $z_{t:t+k}$. This reconstruction task ensures that the learned latent actions $z_{t:t+k}$ contain sufficient information to explain scene dynamics. The model is optimized using three complementary loss components: pixel-level $L_1$ reconstruction loss $\mathcal{L}_{\text{rec}}$ for accurate frame prediction, perceptual loss $\mathcal{L}_{\text{LPIPS}}$ (Zhang et al., 2018) to capture high-level structure beyond pixel accuracy, and optical flow consistency loss $\mathcal{L}_{\text{flow}}$ to encourage physically plausible motion patterns:

$$\mathcal{L}_{\text{flow}} = \frac{1}{L} \sum_{k=t+1}^{t+L} \|\text{OF}(\hat{o}_k, \hat{o}_{k-1}) - \text{OF}(o_k, o_{k-1})\|_1 + \frac{1}{L} \sum_{k=t}^{t+L-1} \|\text{OF}(\hat{o}_k, \hat{o}_{k+1}) - \text{OF}(o_k, o_{k+1})\|_1$$

(1)

where $\text{OF}(a, b)$ denotes optical flow between frames $a$ and $b$ computed via RAFT (Teed & Deng, 2020). This loss encourages predicted frames to exhibit motion patterns consistent with ground truth, supporting temporally coherent dynamics. The total latent action learning loss combines all components:

$$\mathcal{L}_{\text{latent}} = \mathcal{L}_{\text{rec}} + \lambda_{\text{LPIPS}}\mathcal{L}_{\text{LPIPS}} + \mathbb{I}(\text{step} > \alpha_{\text{flow}})\lambda_{\text{flow}}\mathcal{L}_{\text{flow}},$$

(2)

where the flow loss is activated only after $\alpha_{\text{flow}}$ warm-up steps to avoid instability from poor early reconstructions. Thus, latent actions serve as a representation of scene dynamics effectively bridging between visual observations and any embodiment-specific control commands. We provide further architecture details in Appendix B.

## 4.2 PRETRAINING MULTIMODAL VIDEO MODEL WITH LATENT ACTIONS

After obtaining discrete latent actions, we design a pretraining scheme leveraging a powerful multimodal video prediction model. Such models are trained on large-scale datasets to jointly reconstruct video tokens and predict aligned language captions, thereby encoding rich semantic cues and dynamic priors about how the world changes in their latent space. By aligning our discrete latent actions with the outputs of the generative model, we can effectively pretrain a *high-level controller* that can learn from video clips.

To this end, we *jointly* predict future visual tokens and latent actions, unifying dynamic scene understanding and abstract control representation in a temporally coherent latent space. Given the most recent observations $(o_{t-1}, o_t)$ and a task description $c$, the model predicts a future frame $o_{t+H}$ that is $H$ steps ahead, along with a latent action sequence $z_{t:t+H-1} = [z_t, z_{t+1}, \ldots, z_{t+H-1}]$ representing the transitions leading to $o_{t+H}$. This multi-step horizon encourages meaningful and distinct scene changes, providing robust conditioning for downstream action inference.

As shown in Fig. 2 (center), we build on the instruction-tuned LWM-Chat-1M (Liu et al., 2024) as our base policy $G_\theta$. We use LWM's VQ-VAE encoder $E_{\text{VQ}}$ to tokenize each frame $o_k$ into discrete visual tokens $x_k = E_{\text{VQ}}(o_k)$. We also extend it with two modules for latent action modeling: (i) a **Latent Action Embedding** head $E_\phi$ that maps each discrete latent token $z_t^i \in \mathcal{C}$ to a $d_z$-dimensional vector $\tilde{z}_t^i = E_\phi(z_t^i)$ in the model's token space, and (ii) a **Latent Action Token Decoder** $H_\psi$, a multi-layer perceptron (MLP) that autoregressively predicts the next latent token $\hat{z}_t^{i+1} = H_\psi\left(G_\theta\left(c, x_{t-1}, x_t, x_{t+H}, \tilde{z}_{<t}, \tilde{z}_t^{\leq i}\right)\right)$ from the transformer hidden state till position $i$. This allows the model to generate latent action sequences in the same autoregressive manner as language or video tokens, leveraging the multimodal token space learned during pretraining.

During training, we apply *teacher forcing* to both visual and latent action predictions. Given tokenized observations $x_{t-1}$ and $x_t$, the model autoregressively predicts the future visual tokens $\hat{x}_{t+H} = G_\theta(c, x_{t-1}, x_t)$. The ground-truth tokens $x_{t+H} = E_{\text{VQ}}(o_{t+H})$ serve as supervision targets for visual prediction. The pretraining objective $\mathcal{L}_{\text{pretrain}}$ combines both components as:

$$\mathcal{L}_{\text{pretrain}} = \underbrace{\sum_{i=1}^{N_{\text{tokens}}} \text{CE}(\hat{x}_{t+H}^i, x_{t+H}^i)}_{\mathcal{L}_{\text{img}}} + \underbrace{\sum_{k=t}^{t+H-1} \sum_{i=1}^{N_{\text{latent}}} \text{CE}(\hat{z}_k^i, z_k^i)}_{\mathcal{L}_{\text{act}}},$$

(3)

where $\text{CE}(a, b)$ denotes the standard cross-entropy loss between logits for $a$ and label $b$.

### 4.3 CONTINUOUS ADAPTATION

While the latent action pretrained video model provides robust semantic grounded physical priors, it lacks the physical precision needed for smooth, low-level robot control. To address this gap, we augment the pretrained model to output continuous actions, utilizing a flow matching decoder trained on real robot trajectories. This adaptation enables temporally smooth, physically consistent motor commands conditioned on visual and linguistic contexts.

As shown in Fig. 2 (right), we augment the video model $G_\theta$ with two action-specific components: (i) an **Action Encoder** $E_\gamma$, and (ii) a **Flow Decoder** $H_\eta$. The encoder $E_\gamma$ embeds continuous noisy actions $u_s \in \mathbb{R}^{H \times D}$ into the token space, while the decoder $H_\eta$ predicts a flow field over the action chunk. Following the flow matching framework from Eq. 7, we sample a target action sequence $a_{t:t+H-1} \in \mathbb{R}^{H \times D}$, draw a noise sample $u_0 \sim \mathcal{N}(0, I)$, and interpolate:

$$u_s = s \cdot u_0 + (1-s) \cdot a_{t:t+H-1}, \quad s \sim \text{Beta}(a, b).$$

We use $a = 1.5$ and $b = 1$ for sampling from Beta distribution. This noisy input is encoded via $f_s = E_\gamma(u_s, s)$, and passed into the transformer along with VQ-encoded image tokens $(x_{t-1}, x_t)$ and language prompt $c$. The model predicts a flow field $\hat{g} = H_\eta(G_\theta(c, x_{t-1}, x_t, f_s))$, which is supervised using the flow matching objective from Eq. 9:

$$\mathcal{L}_{\text{FM}} = \|a_{t:t+H-1} - u_0 - (1-s) \cdot \hat{g}\|_2^2.$$

At inference time, given the visual history $(o_{t-1}, o_t)$ and task instruction $c$, we iteratively solve for the continuous action chunk $a_{t:t+H-1}$ using forward Euler integration (Eq. 10) of the predicted flow field from $s = 0$ to $s = 1$, over 10 uniform steps with $\Delta s = 0.1$. This continuous control refinement layer injects dynamics consistency and smoothness unavailable to the discrete latent tokens alone.

## 5 EXPERIMENTS

To evaluate ViPRA, we conduct extensive experiments in both simulation and the real world to address the following research questions: (i) Can a generalist policy be trained to leverage both the physical dynamics and semantic understanding of video models? (ii) Does ViPRA efficiently exploit large-scale, actionless video data? (iii) Can multimodal pretraining yield strong high-level priors for downstream policy? (iv) How well does ViPRA adapt to high-frequency continuous control settings? (v) Does ViPRA outperform methods that do not exploit video foundation models?

### 5.1 ENVIRONMENTS & TRAINING

**Training Dataset** For learning latent actions and pretraining the video-language model, we use 198k human videos from Something-Something v2 (Goyal et al., 2017) and a subset of actionless robot videos from the OpenX (Collaboration et al., 2023) dataset: 87k Fractal (Brohan et al., 2022) videos, 25.4k BridgeV2 (Ebert et al., 2021), and 85.6k Kuka (Kalashnikov et al., 2018) videos. We describe training details and hyperparameters for latent action learning in Appendix B, pretraining in Appendix C, and flow matching finetuning in Appendix D.

**Simulation Benchmarks** Following prior latent action works (Ye et al., 2024; Bu et al., 2025), we benchmark ViPRA in SIMPLER (Li et al., 2024b), an open-source suite for evaluating generalist manipulation policies. We evaluate on four Bridge task with a 7-DoF WidowX arm, a benchmark designed to test generalization across diverse manipulation goals. Since SIMPLER lacks finetuning data, we collect 100 diverse multi-task trajectories using a pretrained VLA model (Ye et al., 2024). We provide details about our SIMPLER tasks and LIBERO Long benchmarks in Appendix E.

**Real World Manipulation** While SIMPLER already provides a strong correlation between simulated and real world policy performance, we further strengthen our findings with rigorous evaluations on physical robots. We evaluate ViPRA on a bimanual setup with two 7-DoF Franka Panda robots. For single-arm experiments, we finetune on three multi-instruction tasks: (1) `pick up cloth and cover` $\langle$object$\rangle$, (2) `pick up` $\langle$object$_1\rangle$ `and place on` $\langle$object$_2\rangle$, and (3) `pick up` $\langle$color$_1\rangle$ `cup and stack on` $\langle$color$_2\rangle$ `cup`. We use GELLO (Wu et al., 2024b) teleoperation to collect 180 trajectories, per task spanning 5 cup colors and 10 object types. For both

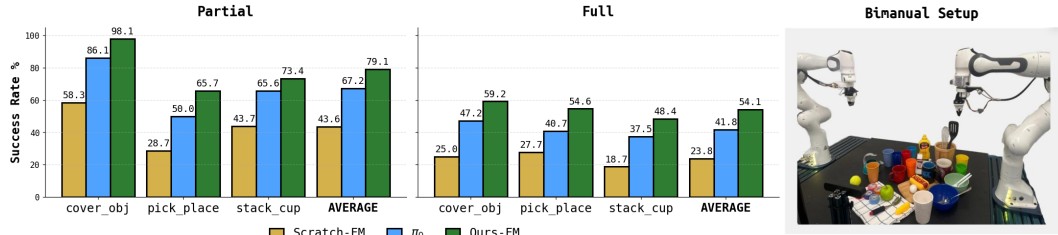

Figure 3: **Real World Evaluations** (Left) We report full and partial success rates on three manipulation tasks. ViPRA-FM significantly outperforms baselines. (Right) We show our physical robot setup and task objects.

| Task | Discrete Actions | | | | | Continuous Actions | | | | |
|---|---|---|---|---|---|---|---|---|---|---|
| | Scratch-AR | VPT | OpenVLA | LAPA | ViPRA-AR | Scratch-FM | UniPI | $\pi_0$ | UniVLA | ViPRA-FM |
| | | | | *Success Rates* | | | | | | |
| StackG2Y | 54.2 | 45.8 | 25.0 | 33.3 | **66.7** | 16.7 | 2.7 | 0.0 | - | **54.2** |
| Carrot2Plate | 58.3 | 37.5 | 20.8 | 41.7 | **62.5** | 33.3 | 2.7 | 20.8 | - | **50.0** |
| Spoon2Cloth | 37.5 | **70.8** | 50.0 | 66.7 | 66.7 | 50.0 | 0.0 | 4.17 | - | **66.7** |
| Eggplant2Bask | 58.3 | 50.0 | 58.3 | 70.8 | **83.3** | 66.7 | 0.0 | 83.3 | - | 79.2 |
| AVG | 52.1 | 51.0 | 38.6 | 53.1 | **69.8** | 41.7 | 1.7 | 27.1 | 42.7 | **62.5** |
| | | | | *Grasp Rates* | | | | | | |
| StackG2Y | 62.5 | 62.5 | **70.8** | 66.7 | 66.7 | 45.8 | 20.8 | 12.5 | - | **62.5** |
| Carrot2Plate | 54.2 | 54.2 | 37.5 | **62.5** | **62.5** | 45.8 | 33.2 | 25.0 | - | **54.2** |
| Spoon2Cloth | 75.0 | 79.2 | 75.0 | **87.5** | 75.0 | 62.5 | 22.2 | 16.7 | - | **79.2** |
| Eggplant2Bask | 66.7 | 70.8 | 91.7 | 79.2 | **100** | 87.5 | 16.0 | 91.7 | - | 91.7 |
| AVG | 65.6 | 66.7 | 68.8 | 73.9 | **76.1** | 60.4 | 23.1 | 36.5 | 50.0 | **71.9** |

Table 1: We report success rates and grasp rates on four bridge tasks in SIMPLER benchmark suite.

simulation and real world settings, we report full success and partial success; partial success is defined as grasping the correct object, and full success requires completing the task (e.g., placing, stacking, covering). We evaluate with both seen and unseen objects, textures, colors, and shapes to test generalization. For real world evaluation, policies run using only a front-facing camera. We predict action chunks of length $H{=}14$ and replan after executing the first 7 steps. For this evaluation, we cap our policies at an effective closed-loop control rate of 3.5 Hz, though they can also operate at higher frequencies upto 22 Hz.

## 5.2 BASELINES

We evaluate ViPRA against strong baselines across discrete and continuous action formulations.

**Scratch.** As a reference, Scratch finetunes the LWM (Liu et al., 2024) video-language backbone directly on downstream tasks with image history and action chunking, without any pretraining. It establishes baseline performance when no latent action or video-based pretraining is used.

**VLA baselines.** We include OpenVLA (Kim et al., 2024) and $\pi_0$ (Black et al., 2024). OpenVLA discretizes actions and adds a one-step autoregressive (AR) action predictor on top of a Prismatic-7B (Karamcheti et al., 2024), while $\pi_0$ augments a PaliGemma-3B (Beyer et al., 2024) with a chunked flow matching (FM) decoder. Both use action-labeled robot demos from OpenX (Collaboration et al., 2023) containing 970k trajectories, while $\pi_0$ also uses proprietary robot data.

**Latent action baselines.** We include LAPA (Ye et al., 2024) and UniVLA (Bu et al., 2025), both of which learn one-step temporally coarse latent actions without video prediction during pretraining. UniVLA improves upon LAPA by learning language-conditioned task-centric actions in DINOv2 (Oquab et al., 2023) space. UniVLA uses a Prismatic-7B backbone with a L1 action decoder, whereas LAPA uses an LWM backbone with one-step AR prediction. Both rely on OpenX demos, with UniVLA additionally leveraging Ego4D (Grauman et al., 2022) and GNM (Yang et al., 2024b).

**Video learning baselines.** We include UniPI (Du et al., 2023b) and VPT (Baker et al., 2022), both of which leverage videos for pretraining. UniPI trains a video diffusion model and trains an inverse dynamics model (IDM) to recovers actions, while VPT trains an IDM on labeled data to extract pseudo-actions that are then used to pretrain an LWM backbone. Reported results are from (Ye et al., 2024), which evaluated them on SIMPLER in a comparable setting.

We include both ViPRA-AR, aligned with discrete autoregressive baselines, and ViPRA-FM, aligned with continuous flow matching methods.

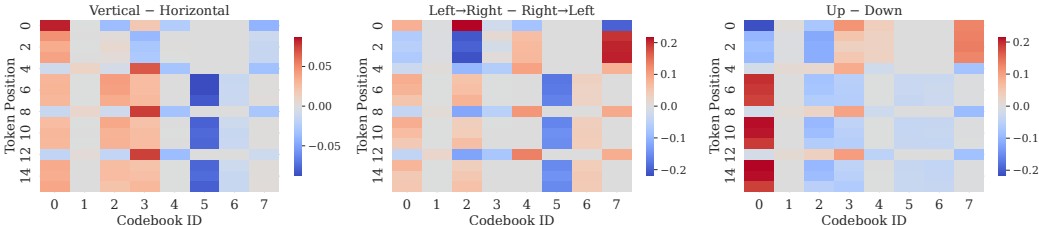

Figure 4: **Positional codebook usage** differences across action categories. Each heatmap shows the difference in per-position token usage between two groups: (left) vertical vs. horizontal, (middle) left → right vs. right → left, and (right) up vs. down. ViPRA learns positionally sensitive codes, with certain entries (e.g., 0, 2, 5) showing systematic variation, indicating that both token index and positions encode action dynamics.

## 5.3 SIMULATION RESULTS

Table 1 shows that ViPRA achieves the highest average success rate across both discrete and continuous settings. In the **discrete setting**, ViPRA-AR surpasses LAPA and OpenVLA by a large margin (69.8% vs. 53.1% and 38.6%), with particularly strong performance on precision-heavy tasks such as `StackG2Y`. These results indicate that temporally fine-grained latent actions improve contact-sensitive manipulation. In the **continuous setting**, ViPRA-FM outperforms Scratch-FM by 20.8%, $\pi_0$ by 35.4%, and UniVLA by 19.8%. This suggests that motion-centric latent pretraining combined with joint future-state modeling provides stronger control priors than both training from scratch and temporally coarse latent approaches. Although ViPRA-AR slightly exceeds ViPRA-FM in the low-noise simulation regime, likely due to faster convergence, ViPRA-FM remains competitive and surpasses all other continuous and discrete baselines. ViPRA also surpasses other video learning, *i.e.*, UniPI and VPT. UniPI frequently generates action sequences that diverge from the given instruction in longer-horizon settings, while VPT provides only limited gains, indicating that IDM-derived pseudo-labels are sensitive to environment shifts. In contrast, ViPRA's joint use of latent action prediction and future state modeling yields stronger cross-environment transfer and more reliable task execution.

## 5.4 REAL WORLD RESULTS

Figure 3 reports performance on three real world manipulation tasks. ViPRA-FM achieves the highest average success rate at 54.1%, outperforming $\pi_0$ (40.1%) and Scratch-FM (23.8%). Notably, this performance is obtained using substantially fewer labeled demonstrations, indicating that dynamics priors learned from actionless videos translate effectively to high-frequency continuous control. We also observe robust retry behavior: after failed grasps, the policy re-attempts interaction, leading to high partial success rates, particularly in `Cover-Obj`, where the cloth is reliably grasped even when placement remains challenging. We exclude discrete policies from real-world evaluation, as bin-based action predictions exhibited unstable spikes under physical noise and occasionally triggered safety stops on the Franka arm. In Appendix F, we provide a detailed comparison of action trajectories across discrete and continuous policies, analyzing the effects of quantization and loss formulations on smoothness and deployment behavior. We further report robustness and generalization analyses in Appendix G, along with evaluations on challenging bimanual tasks in Appendix H. Although the pretraining corpus contains only single-arm robot clips and human videos, ViPRA transfers effectively to coordinated dual-arm settings, suggesting that motion-centric latent pretraining provides embodiment-agnostic priors that adapt with minimal finetuning.

## 5.5 ABLATIONS & ANALYSIS

**Isolating effect of future state and latent prediction.** Table 2 disentangles the contributions of future state prediction and latent action chunking. The LAPA baseline (latent-only) reaches 53.1%, while adding state prediction in ViPRA–*AC* boosts performance to 59.2%, showing that anticipating future observations improves control even with 1-step latents. Removing state prediction from our setup in ViPRA–*SP2* causes a drop from 69.8% to 59.4% (AR) and from 62.5% to 53.2% (FM), underscoring its importance for policy transfer. A state-only variant ViPRA–*LA* achieves 60.7%, comparable to ViPRA–*AC* but still below the full model, indicating that state prediction alone is not sufficient. Finally, adding state prediction at finetuning ViPRA+*SP3* degrades performance (53.1%

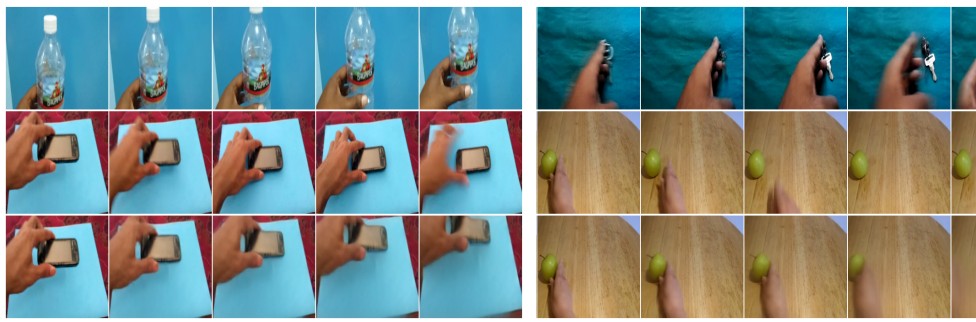

(a) Vertical motion transfer (down → up).  (b) Horizontal motion transfer (right → left).

Figure 5: **Latent action transfer rollouts** illustrating how latent actions extracted from one video can be transferred to a different video. Each column is organized as: **Top** shows the *source clip* $o_{0:4}^A$, used to extract latent actions $z_{0:3}^A = I_\beta(o_{0:4}^A)$ using the inverse model. **Middle** shows the *target clip* $o_{0:4}^B$ with different action which provides a new scene **Bottom** shows the *transferred rollout*, generated by the forward model $o_{0:4}^{B\to A} = F_\alpha(o_0^B, z_{0:3}^A)$, where the latent actions are simulated from the first frame of the target clip. This produces novel behaviors in which the target scene undergoes the source motion.

| Exp. | Pretrain | Finetune | Succ. |
|------|----------|----------|-------|
| LAPA | 1-step L | 1-step A | 53.1 |
| ViPRA-AR | FS + $H$-step L | $H$-step A | **69.8** |
| –AC | FS + 1-step L | 1-step A | 59.2 |
| –SP2 | $H$-step L | $H$-step A | 59.4 |
| –LA | FS only | $H$-step A | 60.7 |
| +SP3 | $H$-step L | FS + $H$-step A | 53.1 |
| ViPRA-FM | FS + $H$-step L | $H$-step A | **62.5** |
| –AC | FS + 1-step L | 1-step A | 44.8 |
| –SP2 | $H$-step L | $H$-step A | 53.2 |
| +SP3 | $H$-step L | FS + $H$-step A | 31.3 |

Table 2: **Ablation study**: Average success rates in SIMPLER on four Bridge tasks, evaluating the contributions of future state prediction (FS), latent action prediction (L), and action chunking (chunk size $H$=14) to justify ViPRA's design choices.

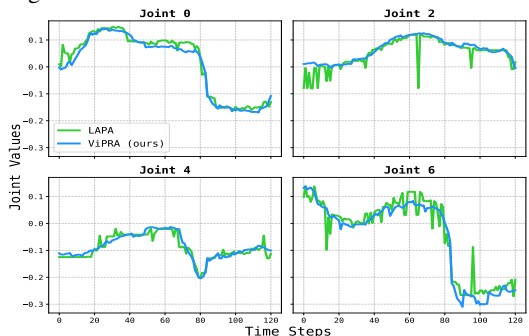

Figure 6: **Action smoothness**: ViPRA-FM (blue) produces smooth, continuous trajectories, while LAPA (green) exhibits local discontinuities and random spikes, often around contact events, despite tracking the overall trend. During real world deployment, such discontinuities triggered the emergency brake mechanism of the robot due to abrupt motor torque jumps.

| Exp. | Succ. | Action MSE |
|------|-------|------------|
| Co-train (Human + Robot) | **0.79** | **0.84** |
| Robot-only | 0.72 | 0.91 |
| Human-only | 0.69 | 0.99 |

Table 3: **Data composition** ablation comparing human-only, robot-only, and co-train on LIBERO-10.

| Exp. | Perplexity | Entropy | Action MSE |
|------|-----------|---------|------------|
| Default (with flow) | **5.01** | **1.59** | **0.84** |
| No flow loss | 5.63 | 1.74 | 0.92 |

Table 4: Ablation on optical flow consistency $\mathcal{L}_{\text{flow}}$ loss during latent action learning. Lower values are better.

AR, 31.3% FM), since the autoregressive structure couples action prediction with video prediction, causing compounding errors that drift irrecoverably on out-of-distribution states. ViPRA mitigates this by jointly predicting future visual tokens and latent action chunks during pretraining only.

**Effect of action chunking.** We apply chunking (Zhao et al., 2023) in both latent and real action spaces: during pretraining the model predicts latent action sequences, and during finetuning it outputs continuous chunks via flow matching. Removing chunking in ViPRA –AC reduces performance to 59.2% (AR) and 44.8% (FM), as single-step actions fail to capture fine temporal dynamics. By combining chunking with future state prediction, the full model achieves the best results of 69.8% (AR) and 62.5% (FM), showing that the two objectives complement each other. Finally, action chunking is not only critical in pretraining but also enables robust, high-frequency control at test time. With KV caching, ViPRA's flow matching decoder runs at 1.95 Hz per chunk, supporting effective control rates up to 22 Hz on hardware (chunk size 14). We provide additional discussion on the connection between action chunking and high-frequency execution in Appendix G.5.

**ViPRA enables smooth continuous control**. To evaluate action smoothness, we compare ViPRA-FM (blue) against LAPA (Ye et al., 2024) (green), a discrete latent policy, during closed-loop rollouts

in Figure 6. Both models are deployed within the same inference pipeline and replayed over trajectories from the finetuning dataset, simulating execution under real visual observations. Although both methods capture the overall motion trend, LAPA exhibits sharp local spikes, particularly at contact events or under partial occlusions, where small perceptual changes induce abrupt bin transitions. In contrast, ViPRA-FM's flow matching decoder produces smooth, demonstration-aligned commands by modeling actions in continuous space. Because such discontinuities can be unsafe on real hardware, we restrict physical robot comparisons to continuous baselines. Additional analysis of discrete and continuous policies is provided in Appendix F, where we examine the effects of quantization, loss design, and action parameterization on control smoothness and deployment stability.

**Latent action analysis**. In Figure 5, a cross-video rollout test illustrates the correlation between latent actions and real action dynamics: injecting latents encoding upward motion from one video (top) into the opening frame of a downward moving video (middle) causes the reconstructions to move upward (bottom), demonstrating transferable, dynamics-aware semantics. Finally, Figure 4 analyzes codebook usage across categories by computing token-position $\times$ code-index histograms. The results indicate that both the choice of codebook entry and its position within the latent action sequence encode structured information about motion direction and dynamics. To further test how motion semantics are shaped during training, we ablate the optical flow consistency loss. As shown in Table 4, removing this supervision increases perplexity and entropy and lowers the accuracy of a action probe with frozen backbone to BridgeV2 actions, confirming that flow alignment encourages the latent space to encode motion rather than background appearance.

**Human and robot data composition ablation**. We repeat latent action learning and pretraining using only robot or only human videos and compare them to our co-training setup. As shown in Table 3, human-only training still yields positive transfer on LIBERO-10, indicating that short-horizon motion cues from human videos remain useful despite the embodiment gap. Robot-only training performs better due to closer kinematic alignment. Co-training gives the best results, with the highest LIBERO-10 success rate and lowest action-probe MSE, suggesting that human videos add motion diversity while robot videos ground the latents in robot dynamics.

# 6 LIMITATIONS AND FUTURE WORK

ViPRA shows that motion-centric latent actions learned from large-scale actionless videos can form a strong foundation for generalist robot policies. Combined with video-language pretraining and flow matching adaptation, this enables smooth, high-frequency control with limited labeled data. While effective across simulation and real robots, several limitations remain and point to important directions for future research.

**Embodiment and dexterity.** Our approach generalizes across embodiments such as WidowX and Franka, including bimanual setups, but extending to more dexterous domains like dual-arm humanoids or multi-fingered hands introduces new challenges. These platforms demand finer latent actions capable of representing precise contact dynamics and inter-arm coordination. Incorporating tactile or force feedback, or learning embodiment-specific adapters, may allow latent actions to better capture such motion primitives.

**Data scope and generalization.** Although ViPRA leverages large-scale actionless videos for pretraining, the diversity of available data remains limited. Expanding to richer ego-centric datasets with human-hand interactions, tool use, or short navigation episodes could improve embodiment invariance and temporal grounding. Integrating additional sensing modalities such as wrist cameras, proprioception, tactile feedback, or depth can further strengthen perception-control alignment, especially in dynamic or unstructured environments.

**Scaling behavior and data efficiency.** An open question is how passive video pretraining scales and where diminishing returns emerge. Characterizing scaling behavior and balancing unlabeled video with smaller quantities of demonstrations would clarify efficient paths for large-scale robot learning.

**Predictive modeling.** The latent action decoder can also be interpreted as a predictive world model. By sampling latent actions, the model can generate visual rollouts, which naturally support reinforcement-learning-based alignment Bai et al. (2022); Rafailov et al. (2023) and test time planning with VLM-based reward models (Ma et al., 2024).

# 7 REPRODUCIBILITY

All models and datasets used in our work are taken from open-sourced components. We take the publicly released LWM-Chat-1M (Liu et al., 2024) as our base video-language model and build on top of that. For latent action learning and pretraining video-language model, we use publicly available datasets: Something-Something (Goyal et al., 2017), Fractal (Brohan et al., 2022), BridgeV2 (Ebert et al., 2021), and Kuka (Kalashnikov et al., 2018). For SIMPLER (Li et al., 2024b) benchmarks, we collected finetuning trajectories by deploying a pretrained VLA model (Ye et al., 2024). We will release this dataset for the community to reproduce our benchmarks. Moreover, we describe training details, hyperparameters, and model architecture for latent action learning in Appendix B, pretraining in Appendix C, and flow matching finetuning in Appendix D. We have released code, models, latent action labeled pretraining data and benchmark scripts at `https://vipra-project.github.io`.

## ACKNOWLEDGMENTS

We thank Jason Liu and Tony Tao for their assistance in conducting the robot experiments, and Jim Yang, Mohan Kumar Srirama, and Tal Daniel for helpful discussions and feedback. This work was supported in part by the Air Force Office of Scientific Research (AFOSR) under Grant No. FA9550-23-1-0747 and by the Office of Naval Research (ONR) MURI under Grant No. N00014-24-1-2748.

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

As part of the supplementary material, we include additional details about the following.

A **Background**: Covers key paradigms relevant for ViPRA: VQ-VAE for learning discrete latent actions, optical flow estimation, behavior cloning for direct action prediction and flow matching for smooth continuous control.

B **Latent Action Learning**: Architecture design, loss formulations, and training protocols for discrete latent action codebook learning, including hyperparameter configurations and optimization strategies.

C **Multimodal Video Pretraining with Latent Actions**: Implementation details for extending LWM with latent actions, including embedding architecture, decoder design, and joint training methodology.

D **Flow Matching Decoder for Continuous Control**: Specification of noise scheduling, action encoder/decoder architectures, and finetuning procedure.

E **Simulation Benchmarks**: Detailed description of SIMPLER tasks and additional LIBERO Long benchmark.

F **Action Output Analysis**: Comparative visualization and discussion of predicted action trajectories across discrete and continuous policies, highlighting the impact of quantization, loss formulations, and control space choices on smoothness and deployment behavior.

G **Real World Experiments**: Detailed description of hardware setup, task design, policy generalization, retrying behavior, and the impact of action chunking on control frequency and real-time performance in physical deployments.

H **Bimanual Manipulation Tasks**: Evaluation of ViPRA-FM on two real world dual-arm tasks requiring spatial coordination and tool use, including task setup, challenges, quantitative results, and rollout visualizations from real robot executions.

We have released code, checkpoints, and latent action labeled data and rollout videos at: `https://vipra-project.github.io`.

# A    BACKGROUND

We review key paradigms relevant for ViPRA: VQ-VAE for imposing information bottleneck to learn discrete latent actions, optical flow estimation, behavior cloning for direct action prediction and flow matching for smooth continuous control.

**Vector-Quantized VAEs (VQ-VAE).** An encoder $e_\zeta$ maps an observation $o_t$ to a continuous latent vector $\tilde{h}_t = [\tilde{h}_t^1, \ldots, \tilde{h}_t^N]$, which is quantized to a sequence of nearest codewords $z_t = [z_t^1, \ldots, z_t^N] \in \mathcal{C}$ using a shared discrete codebook. A decoder $d_\zeta$ reconstructs the observation $\hat{o}_t = d_\psi(z_t + \text{Err})$, where Err is a gradient estimator used to allow backpropagation through the non-differentiable quantization operation.

In traditional VQ-VAE (van den Oord et al., 2017), this estimator takes the form

$$\text{Err}_{\text{STE}} = \text{sg}(\tilde{h}_t - z_t), \tag{4}$$

where $\text{sg}(\cdot)$ denotes the stop-gradient operator. The decoder input $z_t + \text{Err}_{\text{STE}}$ preserves the forward pass while enabling gradients to bypass the non-differentiable argmin. However, this approach typically requires auxiliary losses, such as the codebook and commitment losses, to stabilize training and encourage codebook usage. We adopt the NSVQ formulation (Vali & Bäckström, 2022), which replaces the deterministic STE with a stochastic noise-injected surrogate

$$\text{Err}_{\text{NSVQ}} = \|z_t - \tilde{h}_t\| \cdot \tilde{\epsilon} \tag{5}$$

where $\tilde{\epsilon} = \epsilon/\|\epsilon\|$ and $\epsilon \sim \mathcal{N}(0, I)$. The decoder thus receives $\hat{o}_t = d_\psi(z_t + \text{Err}_{\text{NSVQ}})$. Crucially, NSVQ enables gradients to flow to both the encoder and codebook using only the reconstruction loss, without requiring additional codebook loss terms. The noise-injected gradient estimator, combined with the *unused codebook replacement* technique applied during early training, significantly improves training stability and mitigates codebook collapse, a common issue in VQ-VAE training.

**RAFT based Optical Flow** Given two images, $o_a$ and $o_b$, RAFT (Teed & Deng, 2020) obtains the dense displacement field $\mathbf{f}_{a \rightarrow b} \in \mathbb{R}^{H \times W \times 2}$ that maps each pixel in frame $o_a$ to its location in $o_b$. It first extracts feature maps $\phi(o_a)$, $\phi(o_b) \in \mathbb{R}^{H' \times W' \times d}$ (where $\phi$ is the feature extractor) and builds the all-pair correlation $\mathcal{R}$, with $\mathcal{R}_{ij,kl} = \langle \phi_{ij}(o_a), \phi_{kl}(o_b) \rangle$. The flow prediction is then refined iteratively: $\mathbf{f}^{(k+1)} = \mathbf{f}^{(k)} + \Delta \mathbf{f}^{(k)}(\mathcal{R})$. When applied to temporally close frames in a video, this flow field $f$ can give a good estimate of motion consistency.

**Behavior Cloning (BC)** is a supervised learning paradigm in robotics that learns policies directly from expert demonstrations. Given a dataset $\mathcal{D} = \{(o_t, a_t)\}_{t=1}^T$ of observation-action pairs from expert trajectories, BC trains a parameterized policy $\pi_{\text{BC}}(a_t | o_t; \theta)$ to minimize a distance metric between predicted and ground-truth actions:

$$\min_\theta \mathbb{E}_{(o_t, a_t) \sim \mathcal{D}} \left[ d(\pi_{\text{BC}}(a_t | o_t; \theta), a_t) \right], \tag{6}$$

where $d(\cdot, \cdot)$ is typically the L1 or L2 distance for continuous actions or cross-entropy for discrete actions. This framework has been extended with high-capacity architectures: diffusion models (Chi et al., 2023; Zhao et al., 2023) parameterize $\pi_{\text{BC}}(a_t | o_t; \theta)$ as a denoising process that learns $p(a_t | o_t)$ through iterative refinement, while VLAs leverage pretrained language models (Touvron et al., 2023; Qwen et al., 2024) and visual encoders (Radford et al., 2021; Oquab et al., 2023; Tschannen et al., 2025) as the backbone architecture for $\theta$, enabling multimodal grounding of actions in visual and linguistic contexts.

**Flow Matching** (Lipman et al., 2022) provides an alternative to diffusion models for learning continuous normalizing flows. While diffusion models learn the full denoising process, flow matching directly learns the vector field that transports samples from a source distribution to a target distribution. This approach offers computational advantages for robotics applications where real-time inference is critical.

Flow matching trains a neural network $g_\theta$ to predict the velocity field along a straight-line interpolation path. Given a source sample $x_0$ (typically Gaussian noise) and target sample $x_1$ (e.g., robot actions), the interpolation creates a path:

$$u_s = s \cdot x_0 + (1 - s) \cdot x_1, \quad \text{where} \quad s \in [0, 1] \text{ parameterizes the interpolation} \tag{7}$$

The model learns to predict the velocity field that guides samples along this path:

$$\frac{\partial}{\partial s} u_s = g_\theta(u_s, s|y), \quad \text{where} \quad y \text{ represents conditioning inputs} \tag{8}$$

In robotics applications, $y$ typically includes visual observations and language commands that specify the desired behavior.

The training objective teaches the model to predict the correct velocity by minimizing the difference between predicted and true flow direction:

$$\mathcal{L}_{\text{FM}} = \mathbb{E}_{(y,x_1)\sim\mathcal{D},\, s\sim\mathcal{U}[0,1]} \left\| \underbrace{x_1 - x_0}_{\text{true direction}} - (1-s) \cdot \underbrace{g_\theta(u_s, s \mid y)}_{\text{predicted velocity}} \right\|_2^2. \tag{9}$$

At inference time, samples are generated by integrating the predicted velocity field from noise ($s = 0$) to the target ($s = 1$):

$$u_{s+\Delta s} = u_s + \Delta s \cdot g_\theta(u_s, s|y), \quad \text{where} \quad \Delta s \text{ is the integration step size} \tag{10}$$

This produces the final sample $x_1 \approx u_1$. Forward Euler integration is commonly used due to its efficiency (Black et al., 2024), though more sophisticated solvers like Heun's method or Runge-Kutta can improve stability for high-dimensional control tasks (Kutta, 1901; Runge, 1895). Flow matching has demonstrated superior smoothness and precision compared to direct action prediction, particularly for temporally extended manipulation tasks (Black et al., 2024; NVIDIA et al., 2025b).

# B  LATENT ACTION LEARNING

We detail our latent action learning framework in Algorithm 1, which extracts latent action sequences $z_{t:t+L}$ from video sequences $o_{t:t+L}$ using a combination of reconstruction, perceptual, and optical flow consistency losses. A detailed diagram of this procedure is shown in Figure 7, and the complete training configuration–including model architecture and optimization hyperparameters–is provided in Table 5.

The inverse dynamics encoder maps each frame $o_k$ into a sequence of spatial features using a DINOv2 (Oquab et al., 2023)-initialized backbone. These features are enriched with clip-level context through factorized spatio-temporal attention layers, where the temporal branch employs bidirectional attention to aggregate information across the full sequence. The contextualized features are then discretized via Noise-Substitution Vector Quantization (NSVQ) (Vali & Bäckström, 2022), producing $N_{\text{latent}}$ discrete codes selected from the shared codebook $\mathcal{C}$, which serve as the latent action tokens for $z_k$.

The forward decoder mirrors this architecture with a factorized spatio-temporal transformer, but applies causal temporal attention so that predictions depend only on the past. It jointly attends to the latent action sequence $z_{t:t+k}$ and the observation history $o_{t:t+k}$ to reconstruct the next frame $o_{t+k+1}$. In addition, we integrate the action-conditioning modules proposed by (He et al., 2025) before each spatio-temporal block in the decoder to better align latent action tokens with visual dynamics.

---

**Algorithm 1** Latent Action Learning (Training Step)

---

**Require:** Video clip of $L+1$ observations $o_{t:t+L} \in \mathbb{R}^{(L+1) \times H \times W \times 3}$
**Require:** Hyperparameters: LPIPS weight $\lambda_{\text{LPIPS}}$, Flow weight $\lambda_{\text{flow}}$, Flow start step $\alpha_{\text{flow}}$
**Require:** Codebook $\mathcal{C} \in \mathbb{R}^{K \times D}$ with $K$ codes of dimension $D$

1: *// Inverse model $I_\beta$ comprises of $\tilde{I}_\beta$ and NSVQ module*
2: Compute contextual embeddings: $h_{t:t+L} \leftarrow \tilde{I}_\beta(o_{t:t+L})$
3: **for** $k = 0$ **to** $L - 1$ **do**
4:     Quantize embedding to latent: $z_{t+k} \leftarrow \text{NSVQ}(h_{t+k}, \mathcal{C})$
5:     Decode next frame: $\hat{o}_{t+k+1} \leftarrow F_\alpha(o_{t:t+k}, z_{t:t+k})$
6: **end for**
7:
8: $\mathcal{L}_{\text{rec}} \leftarrow \sum_{k=1}^{L} \|\hat{o}_{t+k} - o_{t+k}\|_1$    *// L1 pixel reconstruction loss*
9: $\mathcal{L}_{\text{LPIPS}} \leftarrow \sum_{t=1}^{L} \text{LPIPS}(\hat{o}_{t+k}, o_{t+k})$    *// Perceptual similarity loss*
10:
11: *// Optical flow consistency loss*
12: **if** step $> \alpha_{\text{flow}}$ **then**
13:     $\mathcal{L}_{\text{flow}} \leftarrow \dfrac{1}{L} \left( \sum_{k=1}^{L} \|\text{OF}(\hat{o}_{t+k}, \hat{o}_{t+k-1}) - \text{OF}(o_{t+k}, o_{t+k-1})\|_1 \right.$
$$\left. + \sum_{k=0}^{L-1} \|\text{OF}(\hat{o}_{t+k}, \hat{o}_{t+k+1}) - \text{OF}(o_{t+k}, o_{t+k+1})\|_1 \right)$$
14: **else**
15:     $\mathcal{L}_{\text{flow}} \leftarrow 0$
16: **end if**
17:
18: $\mathcal{L}_{\text{latent}} \leftarrow \mathcal{L}_{\text{rec}} + \lambda_{\text{LPIPS}} \mathcal{L}_{\text{LPIPS}} + \lambda_{\text{flow}} \mathcal{L}_{\text{flow}}$
19: Update parameters via AdamW optimizer

---

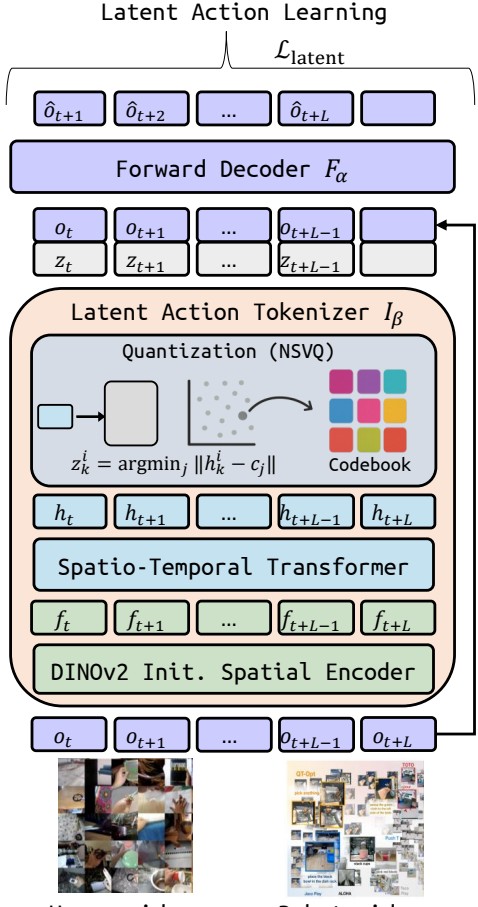

**Figure 7: Latent action learning framework.** Given a sequence of frames $o_{t:t+L}$, an inverse dynamics encoder $I_\beta(z_{t:t+L-1} \mid o_{t:t+L})$ maps observations into discrete latent action sequence via vector quantization. A forward decoder $F_\alpha(\hat{o}_{t+k+1} \mid o_{t:t+k}, z_{t:t+k})$ then reconstructs the future frame conditioned on the observation and latent action history. vTraining combines L1 pixel reconstruction, perceptual (LPIPS), and optical flow consistency losses to ensure that the latent tokens capture physically grounded and temporally localized action information.

| Hyperparameter | Value |
|---|---|
| **Training Configuration** | |
| Optimizer | AdamW |
| Base Learning Rate | 1e-4 |
| DINO Enc. Learning Rate | 1e-5 |
| Optimizer Momentum | $\beta_1, \beta_2 = 0.9, 0.99$ |
| Batch Size | 128 |
| Grad. Norm Clip | 4.0 |
| Total Steps | 240000 |
| Image Augmentation | RandomResizeCrop |
| Flow Start Step $\alpha_{\text{flow}}$ | 60000 |
| Losses | $\mathcal{L}_{\text{rec}} + \lambda_{\text{LPIPS}}\mathcal{L}_{\text{LPIPS}} + \lambda_{\text{flow}}\mathcal{L}_{\text{flow}}$ |
| LPIPS Weight $\lambda_{\text{LPIPS}}$ | 0.5 |
| Flow Weight $\lambda_{\text{flow}}$ | 0.1 (after $\alpha_{\text{flow}}$) |
| GPU | 8 Nvidia H100 (168 hours) |
| **Inverse Dynamics Encoder $I$** | |
| Backbone Init. | DINOv2 (Oquab et al., 2023) |
| Embedding Dim | 768 |
| Spatio-temporal Layers | 6 |
| Attention Heads | 16 |
| Attention Head Dim | 64 |
| **Latent Action Quantization** | |
| Codebook Size $\lvert\mathcal{C}\rvert$ | 8 |
| Quantized Token Dim | 32 |
| Quantization Method | NSVQ (Vali & Bäckström, 2022) |
| Codebook Refresh Interval | Every 10 till 100, every 100 till 1000, every 1000 till 10000 |
| Codebook Refresh Strategy | Re-init Unused, Re-shuffle Used |
| **Forward Decoder $F_\alpha$** | |
| Embedding Dim | 768 |
| Spatio-temporal Layers | 8 |
| Attention Heads | 16 |
| Attention Head Dim | 64 |

Table 5: Hyperparameters for latent action learning.

## C  MULTIMODAL VIDEO PRETRAINING WITH LATENT ACTIONS

We augment LWM-Chat-1M (Liu et al., 2024), a pretrained multimodal video model $G_\theta$ with a latent action embedding layer $E_\phi$ and a latent action decoder $H_\psi$. The model jointly predicts end-of-horizon visual tokens $x_{t+H}$ and latent action sequences $z_{t:t+H-1}$ leading to it, conditioned on history visual tokens $(x_{t-1}, x_t)$ and task context $c$.

The latent action embedding head $E_\phi$ maps each code $z_t^i \in \mathcal{C}$ into the token space of $G_\theta$, and the decoder head $H_\psi$ predicts next-token logits over the latent vocabulary. Training uses teacher forcing for both visual tokens and latent tokens with cross-entropy loss. The complete training procedure and hyperparameters are detailed in Algorithm 2 and Table 6.

---

**Algorithm 2** Pretraining Multimodal Video Model via Video and Latent Action Prediction

---

**Require:** History frames: $(o_{t-1}, o_t)$
**Require:** Target frame: $o_{t+H}$
**Require:** Task description: $c$   *// Tokenized text string*
**Require:** Target latent action chunk: $z_{t:t+H-1} \in \mathcal{C}^{H \times N_\text{latent}}$
**Require:** Pretrained models: VQ-VAE encoder $E_\text{VQ}$, video model $G_\theta$
**Require:** Trainable components: $G_\theta$, random initialized embedding layer $E_\phi$, decoder head $H_\psi$
 1: Tokenize input frames: $x_{t-1}, x_t \leftarrow E_\text{VQ}(o_{t-1}), E_\text{VQ}(o_t)$
 2: Tokenize target frame: $x_{t+H} \leftarrow E_\text{VQ}(o_{t+H})$
 3: Embed latent actions: $\tilde{z}_{t:t+H-1} \leftarrow E_\phi(z_{t:t+H-1})$
 4:
 5: **for** $i = 1$ **to** $N_\text{tokens}$ **do**
 6:    $\hat{x}_{t+H}^i \leftarrow G_\theta\left(c, x_{t-1}, x_t, x_{t+H}^{<i}\right)$   *// Autoregressive prediction with teacher forcing*
 7: **end for**
 8:
 9: $\mathcal{L}_\text{img} \leftarrow \sum_{i=1}^{N_\text{tokens}} \text{CE}\left(\hat{x}_{t+H}^i, x_{t+H}^i\right)$   *// Image token loss*
10:
11: **for** $k = t$ **to** $t + H - 1$ **do**
12:    **for** $i = 1$ **to** $N_\text{latent}$ **do**
13:       $\hat{z}_k^i \leftarrow H_\psi\left(G_\theta\left(c, x_{t-1}, x_t, x_{t+H}, z_{<k}, z_k^{<i}\right)\right)$
14:    **end for**
15: **end for**
16:
17: $\mathcal{L}_\text{act} \leftarrow \sum_{k=0}^{H-1} \sum_{i=1}^{N_\text{latent}} \text{CE}\left(\hat{z}_{t+k}^i, z_{t+k}^i\right)$   *// Latent action token loss*
18:
19: $\mathcal{L}_\text{pretrain} \leftarrow \mathcal{L}_\text{img} + \mathcal{L}_\text{act}$   *// Total loss*
20: Update $G_\theta$, $E_\phi$, and $H_\psi$ using AdamW optimizer

---

| Hyperparameter | Value |
|---|---|
| **Model Setup** | |
| Video Model | LWM-Chat-1M (Liu et al., 2024) (initialized) |
| Tokenization Backbone | VQ-VAE (frozen) |
| Prompt Tokenizer | BPE tokenizer |
| Latent Action Vocabulary Size $|\mathcal{C}|$ | 8 |
| Latent Embedding Dim ($E_\phi$) | 4096 |
| Latent Decoder Type ($H_\psi$) | MLP |
| Latent Decoder Layers | 1 |
| Latent Decoder Hidden Dim | 2048 |
| **Training Configuration** | |
| Optimizer | AdamW |
| Learning Rate | 4e-5 |
| Weight Decay | 0.0 |
| Optimizer Betas | $(0.9, 0.95)$ |
| Batch Size | 512 |
| Total Steps | 50,000 |
| Dropout | 0.1 |
| Gradient Clipping | 1.0 |
| Mixed Precision | `bfloat16` |
| GPU | 8 Nvidia H100 (144 hours) |
| **Prediction Targets** | |
| Prediction Horizon $H$ | 14 |
| Image Token Loss | Cross Entropy |
| Latent Action Loss | Cross Entropy |

Table 6: Hyperparameters for multimodal video pretraining to jointly predict future visual state and latent action sequence. The video model is initialized from LWM-Chat-1M and trained jointly with lightweight latent action modules.

## D FLOW MATCHING DECODER FOR CONTINUOUS CONTROL

To enable continuous control, we augment the pretrained video model $G_\theta$ with a flow-based action decoder. Specifically, we introduce (i) a noisy action encoder $E_\gamma$ that embeds a time-interpolated noisy action chunk $u_s$ into the embedding space of the model, and (ii) a flow decoder $H_\eta$ that predicts the velocity field used to transport the noisy actions toward the ground-truth action chunk $a_{t:t+H-1}$. The full training procedure is described in Algorithm 3, and architectural details and hyperparameters are provided in Table 7.

---

**Algorithm 3** Flow Matching for Continuous Control

---

**Require:** Image history frames $(o_{t-1}, o_t)$,
**Require:** Task description: $c$     // *Tokenized text string*
**Require:** Action chunk $a_{t:t+H-1} \in \mathbb{R}^{H \times D}$     // *D-dimensional actions H-steps into the future*
**Require:** Pretrained models: VQ-VAE encoder $E_{\text{VQ}}$, video model $G_\theta$
**Require:** Trainable components: $G_\theta$, noisy action encoder $E_\gamma$, flow decoder $H_\eta$
 1: Sample timestep $s \sim \text{Beta}(1.5, 1.0)$
 2: Sample noise $u_0 \sim \mathcal{N}(0, I)$
 3: Compute interpolation: $u_s \leftarrow s \cdot u_0 + (1-s) \cdot a_{t:t+H-1}$
 4: Tokenize input frames: $x_{t-1}, x_t \leftarrow E_{\text{VQ}}(o_{t-1}), E_{\text{VQ}}(o_t)$
 5: Encode noisy actions: $f_s \leftarrow E_\gamma(x_s, s)$
 6: Predict flow field: $\hat{g} \leftarrow H_\eta\left(G_\theta\left(c, x_{t-1}, x_t, f_s\right)\right)$
 7: $\mathcal{L}_{\text{FM}} \leftarrow \|a_{t:t+H-1} - u_0 - (1-s) \cdot \hat{g}\|_2^2$     // *Flow matching loss*
 8: Update $G_\theta$, $E_\gamma$, $H_\eta$ using AdamW optimizer

---

| Hyperparameter | Value |
|---|---|
| **Noisy Action Encoder Head** ($E_\gamma$) | |
| Architecture | 2-layer MLP |
| Hidden Dim | 4096 |
| Embedding Dim ($d_a$) | 4096 |
| Activation | GELU |
| Dropout | 0.1 |
| **Flow Decoder Head** ($G_\eta$) | |
| Input Dim | 7 (End-Effector Deltas) or 8 (Absolute Joint States) |
| Architecture | Single linear projection |
| **Flow Matching Setup** | |
| Interpolation Timestep $s$ | Beta(1.5, 1.0) |
| Noise Distribution $\mathbf{x}_0$ | Standard normal $\mathcal{N}(0, I)$ |
| Prediction Horizon $H$ | 14 |
| Integration Method (Inference) | Forward Euler, $N = 10$ steps |
| **Training Configuration (follows Table 6)** | |
| Total Steps | 12000 (SIMPLER) |

Table 7: Hyperparameters used for flow matching action decoder to enable continuous control.

# E    SIMULATION BENCHMARKS

## E.1    SIMPLER BENCHMARK

Following prior latent action works (Ye et al., 2024; Bu et al., 2025), we benchmark ViPRA in SIM-PLER (Li et al., 2024b), an open-source suite for evaluating generalist manipulation policies. We evaluate on four Bridge tasks with a 7-DoF WidowX arm, a benchmark designed to test generalization across diverse manipulation goals. Since SIMPLER does not provide finetuning data, we collect 100 diverse multi-task trajectories using a pretrained VLA model (Ye et al., 2024) to adapt policies before evaluation. The tasks are as follows:

- **Spoon2Cloth**: The instruction is `put the spoon on the towel`. The spoon is placed on a vertex of a 15 cm square on the tabletop, and the towel on another vertex. The spoon's orientation alternates between horizontal and vertical, requiring the robot to re-orient its gripper. This task evaluates both grasp selection and orientation adjustment.

- **Carrot2Plate**: The instruction is `put carrot on plate`. Same setup as Spoon2Cloth, but with a carrot and a plate. While similar in layout, this introduces a different geometry and surface, requiring adaptation in grasping and placement.

- **StackG2Y**: The instruction is `stack the green block on the yellow block`. A green block is placed on a vertex of a tabletop square (10 cm and 20 cm edges) and a yellow block on another. Success requires precise alignment and careful release, making it a fine-grained manipulation task that stresses stability and accuracy.

- **Eggplant2Bask**: The instruction is `put eggplant into yellow basket`. An eggplant is dropped into the right basin of a sink and a yellow basket in the left basin. The eggplant is randomized in pose but ensured to be graspable. This task evaluates robustness to shape variability and placement under uncertainty, as the object must be reliably picked and transferred across workspace regions.

We evaluate performance using two metrics: **success rate** and **partial success rate (grasp rate)**. Success rate measures whether the full task goal is completed (e.g., spoon placed on towel, block stacked without falling, eggplant deposited into basket). Grasp rate captures whether the robot is at least able to establish a successful grasp on the object, even if the subsequent placement or stacking is not achieved. This distinction is important: grasping reflects a fundamental capability for initiating manipulation, while successful completion requires the integration of grasping with precise transport and placement. Together, these metrics provide a more comprehensive view of policy competence, distinguishing between failures due to perception/grasping versus those arising from downstream control and placement.

## E.2    SIMPLER RESULTS

We report both end-to-end *success rate* and *grasp rate* in Table 1. Across **discrete actions** setting, **ViPRA-AR** attains the best average success (69.8%), exceeding LAPA (53.1%), VPT (51.0%) and OpenVLA (38.6%). It leads on precision-heavy `StackG2Y` (66.7% vs. 54.2% Scratch-AR, 45.8% VPT) and `Carrot2Plate` (62.5%), and remains competitive on `Spoon2Cloth` (66.7%, near VPT's 70.8%). On `Eggplant2Bask`, ViPRA-AR (83.3%) significantly outperforms other methods, demonstrating strong transport and placement.

In the **continuous setting**, **ViPRA-FM** achieves the highest average success (62.5%), outperforming Scratch-FM (41.7%), $\pi_0$ (27.1%), and UniVLA (42.7%). It is the strongest continuous model on `StackG2Y` (54.2%), `Carrot2Plate` (50.0%) and `Spoon2Cloth` (66.7%) while remaining competitive (79.2%) with $\pi_0$ (83.3%) on `Eggplant2Bask`. UniPI frequently generates action sequences that diverge from the given instruction in longer-horizon settings, while VPT provides only limited gains, indicating that IDM-derived pseudo-labels are sensitive to environment shifts. In contrast, ViPRA's joint use of latent action prediction and future state modeling yields stronger cross-environment transfer and more reliable task execution.

ViPRA converts grasps into task completion more reliably. On `StackG2Y`, OpenVLA achieves 70.8% grasp but only 25.0% success (a 45.8 pt gap), indicating post-grasp placement failures. ViPRA-AR maintains 66.7% grasp and 66.7% success (0 pt gap), and ViPRA-FM 62.5% grasp

| Method | Success Rate |
|---|---|
| UniPI (Du et al., 2023c) | 0.00 |
| OpenVLA Kim et al. (2024) | 0.54 |
| $\pi_0$-FAST (Black et al., 2024) | 0.60 |
| $\pi_0$ (Black et al., 2024) | 0.85 |
| UniVLA (Bu et al., 2025) | 0.92 |
| UVA | 0.90 |
| ViPRA-FM | 0.79 |

Table 8: Success rates on LIBERO-10 benchmark.

vs. 54.2% success (8.3 pt gap), evidencing stable transport and release. On `Eggplant2Bask`, Open-VLA's 91.7% grasp falls to 58.3% success (33.4 pt drop), whereas ViPRA-AR ($100\% \rightarrow 83.3\%$) and ViPRA-FM ($91.7\% \rightarrow 79.2\%$) show markedly smaller drops, consistent with smoother post-grasp control and accurate instruction following.

### E.3 LIBERO LONG BENCHMARK

We also evaluate on LIBERO Long (a.k.a LIBERO 10) (Liu et al., 2023), the most challenging subset of the LIBERO simulation benchmark. Unlike the Spatial, Object, or Goal subsets, LIBERO Long focuses on long-horizon manipulation tasks that require sequencing multiple sub-goals with heterogeneous objects, layouts, and task dependencies. This setting stresses robustness and temporal compositionality, since errors can accumulate across long horizons.

The evaluation consists of a suite of 10 long-horizon tasks, each paired with a natural language goal description. For example, one of the task instructions includes `"put the white mug on the plate and put the chocolate pudding to the right of the plate"`, requiring reasoning over both symbolic relations (object identities, spatial references) and low-level control. For each task, the environment is initialized with objects placed in varied locations, increasing the difficulty of generalization.

Each task is evaluated across 10 runs with 5 different random seeds, and results are reported as the average reward over all 10 tasks (500 episodes in total). This protocol provides a stringent test of both semantic grounding and long-horizon policy execution, making LIBERO Long a valuable complement to SIMPLER's shorter-horizon manipulation tasks.

### E.4 LIBERO LONG RESULTS

On LIBERO-10, which emphasizes long-horizon, multi-stage manipulation, ViPRA achieves a 79% success rate. This is substantially higher than OpenVLA (54%) and $\pi_0$-FAST (60%), and close to UVA (90%), which is specifically optimized for LIBERO. These results demonstrate that ViPRA's motion-centric latent pretraining transfers effectively to simulated long-horizon tasks, outperforming methods trained primarily with labeled actions or direct policy supervision.

We observe that ViPRA performs reliably on coarse manipulations (e.g., cups, bowls, books), which are easy to grasp, but struggles with precision grasps such as cylindrical cans that require diameter-aligned control. We attribute this to *delta-EEF drift*: since LIBERO's action space is delta end-effector, small prediction biases can accumulate over time, leading to imprecise grasps in the absence of absolute cues to re-anchor the trajectory. For instance, $\pi_0$ mitigates this issue by conditioning on proprioceptive state history and wrist-camera inputs. Despite lacking such additional signals, ViPRA surpasses OpenVLA under the same sensing setup (image-only, delta-EEF), underscoring the benefits of motion-centric latent pretraining for long-horizon manipulation.

# F    ACTION OUTPUT ANALYSIS

We provide a more in-depth analysis of the action outputs of various policies introduced in Section 5.5, highlighting their differences in smoothness, consistency, and suitability for real world deployment. Policies differ in their action representations and control spaces:

- **Absolute Joint Space.** ViPRA-FM and LAPA (Ye et al., 2024) output full 7D joint positions (Franka), directly supervised in joint space.
- **Delta End-Effector Space.** OpenVLA (Kim et al., 2024), $\pi_0$ (Black et al., 2024), and operate in 7D Cartesian delta commands (position, Euler rotations, gripper), decoded from visual inputs.
- **Continuous vs Discrete.** ViPRA-FM and $\pi_0$ (Black et al., 2024) predict continuous actions via a flow matching decoder, whereas LAPA (Ye et al., 2024) and OpenVLA (Kim et al., 2024) use quantized logits over discretized action bins.

To better understand the behavioral differences between discrete and continuous policies, we analyze the predicted action trajectories across different models during closed-loop visual rollout on real robot observations. We evaluate policies by loading their finetuned checkpoints into our inference pipeline and simulating replay on the training trajectories from the finetuning dataset. This allows us to visualize their motor command trends without introducing new generalization factors. In particular, we compare ViPRA-FM (ours), LAPA (Ye et al., 2024), OpenVLA (Kim et al., 2024), and $\pi_0$ (Black et al., 2024).

LAPA (Ye et al., 2024) and OpenVLA (Kim et al., 2024) rely on a discretization scheme in which each dimension of the robot's action space is uniformly quantized into 255 bins using equal-sized quantiles over the training distribution. That is, for each joint or end-effector dimension, bin boundaries are chosen so that each bin contains roughly the same number of training points. This quantile-based discretization ensures equal data coverage across bins but introduces two key limitations in how actions are represented and learned:

1. **Contact-Sensitive Flipping:** At test time, small perturbations in the input (e.g., due to occlusions or slight viewpoint drift) may cause the model to flip from one bin to another near the quantile boundary–especially at contact points. Since adjacent bins can correspond to different action magnitudes, these minor visual shifts can lead to abrupt discontinuities in motor output.
2. **Loss Granularity:** The cross-entropy loss used for training treats each action bin as a distinct class label. As a result, all incorrect predictions are penalized equally, regardless of how close they are to the ground-truth bin. For example, predicting bin 127 instead of 128 incurs the same loss as predicting bin 0. This is fundamentally at odds with the structure of continuous action spaces, where the cost of an error should scale with its magnitude.

We hypothesize that the combination of bin boundary flipping and non-metric loss leads to the spiky or erratic behavior seen in discrete action models, particularly around moments of contact or high-frequency motion. These effects are amplified in high-dimensional control settings, where discretization artifacts can arise independently in each action dimension–compounding into visibly unstable or jerky behaviors across the full joint trajectory.

By contrast, continuous policies such as ViPRA-FM and $\pi_0$ (Black et al., 2024) operate directly in $\mathbb{R}^D$ using flow matching. These losses naturally reflect the structure of the action space–penalizing predictions in proportion to how far they deviate from the ground truth. As a result, the output trajectories tend to be smoother, better aligned with demonstrations, and more robust to perceptual jitter.

We note that it may be possible to mitigate some of the above issues by increasing the number of bins or by using non-uniform binning schemes (e.g., higher resolution in frequently visited regions). However, these approaches increase model complexity and still inherit the fundamental limitation of using classification loss in a regression setting. Continuous decoders trained with distance-aware objectives offer a more natural and principled solution for low-level control.

To gain deeper insight into how different action representations influence control behavior, we examine the temporal structure of predicted actions across several rollout trajectories. We organize

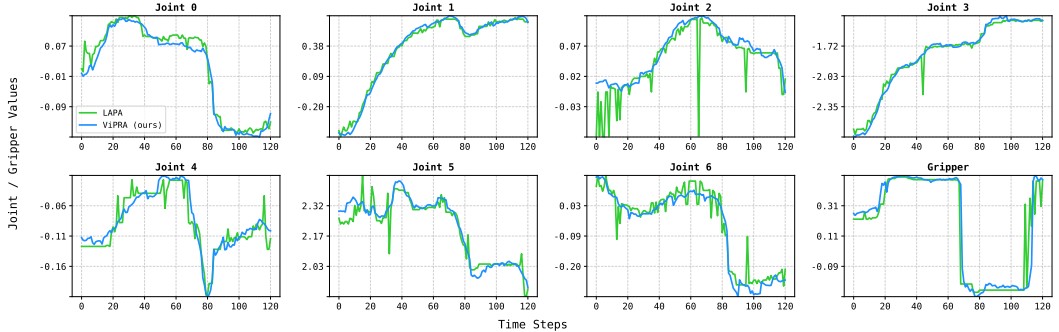

(a) **Absolute joint space.** Predicted 7D joint positions over time for ViPRA-FM (blue) and LAPA (Ye et al., 2024) (green). ViPRA-FM produces smooth, continuous trajectories, while LAPA (Ye et al., 2024) exhibits local discontinuities and random spikes–often around contact events–despite tracking the overall trend. In real world deployment, such discontinuities *triggered Franka's emergency brake mechanism due to abrupt torque jumps.*

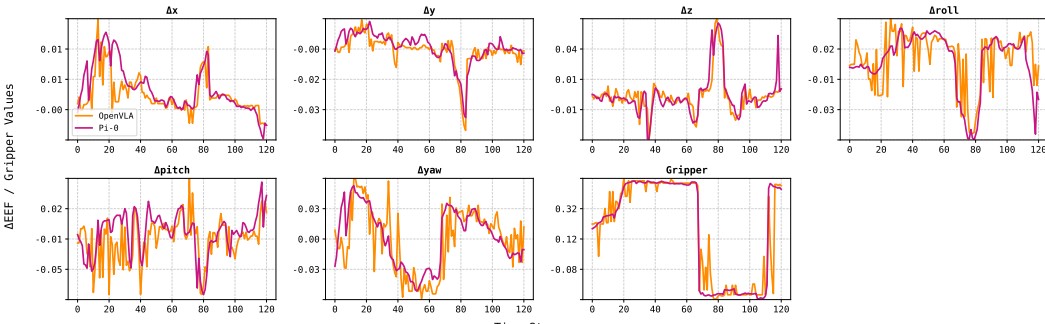

(b) **Delta end-effector space.** Predicted 7D delta actions (position, rotation, gripper) for $\pi_0$ (Black et al., 2024) (magenta) and OpenVLA (Kim et al., 2024) (orange). Although delta control provides structured low-level modulation, OpenVLA exhibits sharp fluctuations due to discretized output. Notably, the gripper signal shows large, momentary switches during contact events–resulting in failed grasps or premature object drops. In contrast, $\pi_0$ maintains stable gripper behavior during fine manipulation.

Figure 8: Visualization of predicted actions across different control spaces. Discrete policies often produce sharp discontinuities due to binning artifacts and classification loss, whereas continuous policies exhibit smoother, dynamics-consistent behavior.

the analysis by control space–absolute joint angles vs. delta end-effector motions–and visualize per-dimension action trends across time in Figure 8.

These visualizations support our hypothesis: *discrete policies, trained with cross-entropy over fixed bins, tend to produce abrupt transitions around perceptually sensitive regions–*especially near bin boundaries or occlusions. This manifests as random spikes, high-frequency jitter, or contact-time instability, all of which can destabilize robot behavior in deployment.

In contrast, *continuous policies like* ViPRA-FM *and* $\pi_0$, *trained with flow matching losses, yield consistently smooth, physically plausible actions that better reflect real world constraints.* The ability to interpolate naturally between states–not just classify them–proves critical for robust closed-loop performance in contact-rich manipulation.

## G    REAL WORLD EXPERIMENTS: SETUP, CHALLENGES, AND OBSERVATIONS

To complement our real world results in Section 5.4, we provide additional details on our hardware setup, task design, and policy behavior under realistic sensing and control constraints. We also analyze generalization to unseen objects, retry behavior, and how chunked continuous actions support efficient closed-loop control.

### G.1    HARDWARE AND DATA COLLECTION SETUP

All experiments are conducted on a real world robotic platform with two 7-DOF Franka Emika Panda arms. The workspace is observed by a single front-mounted ZED stereo camera. There are no wrist-mounted or side-view cameras, so all perception is monocular and from a fixed third-person viewpoint. We use the GELLO teleoperation system (Wu et al., 2024b) to collect human demonstrations at 15Hz. Demonstrations are collected directly in task-relevant environments, with each policy trained using only a single camera view.

Our decision to use image history as part of the observation is motivated by the inherently temporal nature of the video model architecture, as well as the absence of auxiliary views. Stacking observations over time allows the model to internally infer dynamics and compensate for occlusions or ambiguous single-frame cues.

### G.2    TASK DESCRIPTIONS AND CHALLENGES

We evaluate policies on three real world single-arm tasks, each with unique control and perception challenges (Figure 9):

1. **Cover-Object:** The robot must pick up a piece of cloth and drape it over a specified object. This task is challenging due to the deformable nature of cloth, which requires reliable grasping from the table surface. Slight changes in cloth configuration or object geometry can affect dynamics drastically. Generalization requires reasoning over unseen cloth textures and novel target objects.

2. **Pick-Place:** The robot must pick up a named object (e.g., sponge, bowl, duck) and place it on a destination surface (plate or board). Object shapes vary significantly, leading to different grasp affordances. Grasping a wide bowl vs. a narrow-handled cup requires distinct motor strategies. The task is highly multimodal–there are multiple correct ways to perform the task, depending on object shape, pose, and placement surface.

3. **Stack-Cups:** The robot must follow language instructions to stack a cup of color1 onto a cup of color2. Success requires grounding object properties and executing precise stacking. Evaluation setups include unseen cup types, color shades, and geometries to test language understanding and spatial generalization.

### G.3    GENERALIZATION TO NOVEL OBJECTS

A core goal of our real world evaluation is to assess how well the policy generalizes to unseen object instances and configurations not encountered during training. We design test-time setups that introduce meaningful variation across tasks:

- **Cover-Object:** Test scenarios include cloths of varying texture, size, and stiffness, as well as new target objects such as jars, boxes, and toys. These variations require the policy to generalize grasp strategies and adapt to deformable material dynamics.

- **Pick-Place:** We evaluate on previously unseen objects with diverse geometries and affordances (e.g., bowls, mugs, fruits), and destination surfaces of varying size and texture. The task requires flexible grasping and reliable placement across a range of object shapes and destination surfaces.

- **Stack-Cups:** Evaluation includes new cup types with unseen shapes, rim sizes, and fine-grained color variations. The policy must generalize language grounding to new color references and execute precise stacking across novel physical configurations.

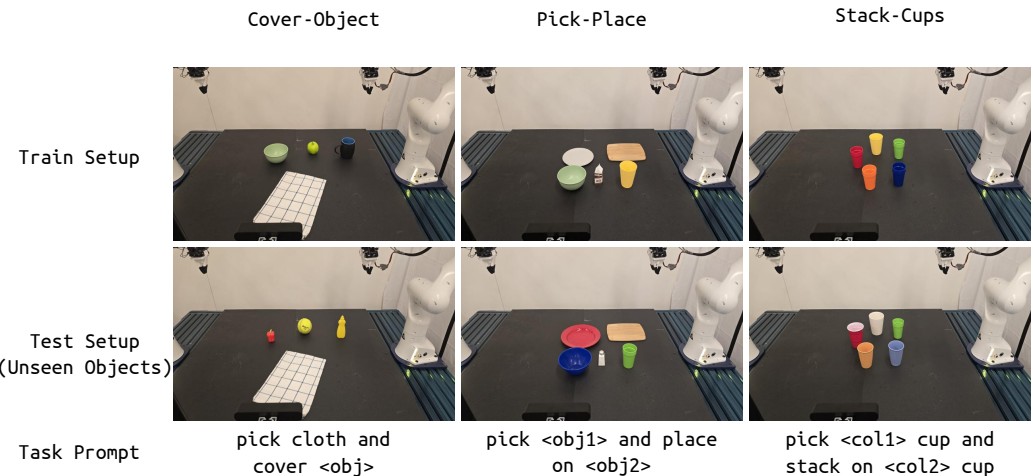

Figure 9: **Task Setup Overview.** (Top row) Training environments for each of the three single-arm manipulation tasks: `Cover-Object`, `Pick-Place`, and `Stack-Cups`. (Bottom row) Evaluation environments featuring novel objects, textures, or placements not seen during training. Note the variety in cloth shape, object geometry, plate type, and cup color/size combinations.

Despite these shifts, *our method consistently exhibits robust generalization across all tasks*. We attribute this to the combination of latent dynamics pretraining, language-conditioned perception, and a unified architecture that integrates semantic, spatial, and temporal cues. Pretraining on diverse unlabeled videos teaches the model general priors about object motion and interaction. Conditioning on task instructions guides object selection and interpretation even in ambiguous or unfamiliar contexts. Finally, the architectural design ensures that learned representations capture not just appearance, but how objects behave across time, enabling transfer to new instances that were not explicitly seen during supervised finetuning.

### G.4 RETRYING BEHAVIOR ENABLED BY TEMPORAL PRETRAINING

Our method consistently *exhibits robust retry behavior*: when an initial grasp attempt fails, due to occlusion, misalignment, or object shift, the policy often reattempts until successful. This is especially evident in `Cover-Object`, where the robot frequently retries grasping if the cloth slips, and in `Pick-Place`, where wide or irregularly shaped objects like bowls may require multiple grasp attempts from different angles.

We attribute this robustness to our temporal pretraining objective. By learning to predict future video frames and latent actions over multiple steps, the model develops a sense of longer-horizon dynamics and recoverability. Rather than depending on single-step feedback, it implicitly plans through extended temporal context–enabling it to course, correct and persist through partial failures.

### G.5 ACTION CHUNKING AND INFERENCE EFFICIENCY

ViPRA produces continuous actions using a chunked flow matching decoder, generating sequences of 14 actions per inference step. At test time, we cap control frequency by evaluating two rollout strategies: **7/14 rollout**, where the first 7 actions of each chunk are executed before re-planning, and **14/14 rollout**, where all 14 actions are executed before the next inference. The former corresponds to an effective closed-loop update rate of ∼3.5 Hz, while the latter doubles this to 7 Hz. Because predicted action trajectories are smooth and temporally coherent, ViPRA remains stable even under open-loop execution within each chunk. This property is particularly beneficial for contact-rich phases that demand reactive yet jitter-free behavior.

**KV caching for fast inference** We further optimize inference with key-value (KV) caching. Language and image attention states are cached once and reused across flow matching Euler steps, so only action tokens are recomputed during integration. This reduces redundant computation, enabling the entire 14-step chunk to be produced in 510 ms (∼1.95 Hz), which corresponds to a robot-side

control frequency of up to 22 Hz. Our setup can stably support control rates approaching 20 Hz, to our knowledge matched only by one other 7B-parameter model (Kim et al., 2025).

**Comparison with baselines.** Table 9 summarizes model sizes, action rollout lengths, and inference times. Unlike prior approaches that also use a 7B model (e.g., LAPA and OpenVLA) and operate at ∼200 ms per step but predict only single actions, ViPRA amortizes inference across long, smooth action chunks, enabling high frequency reactive control.

| Method | Model Size | Action Steps | Inference Time (ms) |
|---|---|---|---|
| LAPA (Ye et al., 2024) | 7B | 1 | 220 |
| OpenVLA (Kim et al., 2024) | 7B | 1 | 190 |
| $\pi_0$ (Black et al., 2024) | 3.3B | 16 | 90 |
| UniPI (Du et al., 2023b) | – | 16 | 24000 |
| UVA (Li et al., 2025a) | 0.5B | 16 | 230 |
| ViPRA (ViPRA-FM) | 7B | 14 | 510 |

Table 9: Inference speed comparison across models. ViPRA achieves high effective control frequencies by amortizing computation over action chunks.

# H ViPRA-FM on Challenging Bimanual Tasks

Bimanual manipulation introduces significant complexity beyond single-arm control. The combined action space spans 14 degrees of freedom, and inter-arm coordination requires precise spatial alignment, collision avoidance, and timing consistency. The solution space is also highly multimodal–there are many valid ways to execute a task depending on object geometry, initial configurations, and movement variability. These challenges make bimanual tasks a strong test of a policy's ability to generalize and coordinate under real world constraints.

## H.1 Bimanual Setup

We test our framework using both arms of the Franka Panda robot. While only the right arm performs active grasping, both arms are controlled jointly using a single policy conditioned on shared language instructions. The system receives monocular observations from a front-mounted ZED camera and generates chunked continuous actions for both arms in a synchronized control loop.

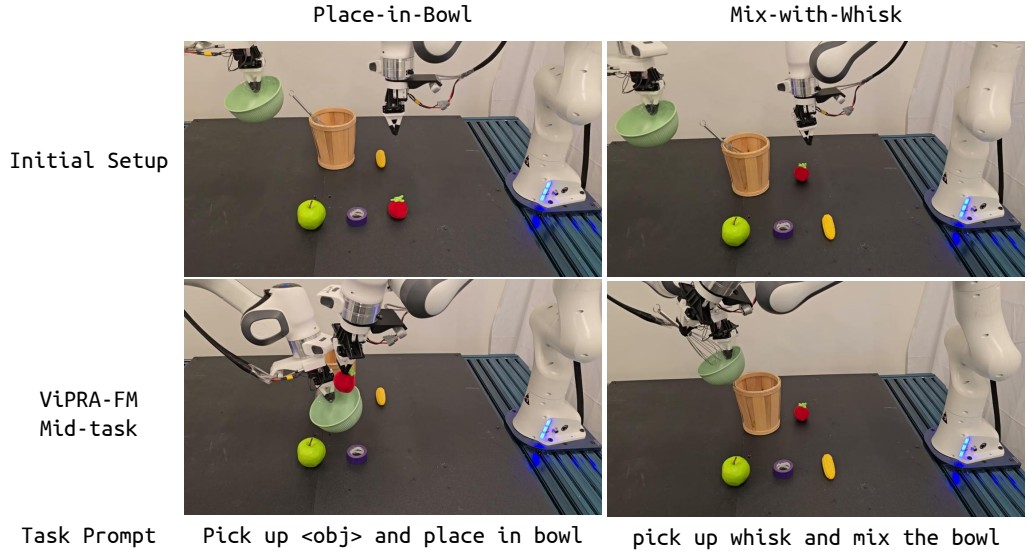

Figure 10: **Bimanual task execution by ViPRA-FM.** (Top row) Initial setup for the two tasks: placing a tomato into a bowl and mixing with a whisk. (Bottom row) Mid-execution rollout of ViPRA-FM: the right arm transports the tomato toward the bowl held by the left arm (left), and mixes the contents using the whisk while the left arm maintains bowl stability (right). These examples highlight coordinated two-arm control and fluent execution of tool- and object-handling behaviors.

We evaluate two bimanual tasks of increasing complexity:

**(1) Place-in-Bowl:** The right arm must grasp a target object (e.g., a fruit or kitchen item) and place it into a bowl held by the left arm. Success requires fine-grained spatial alignment above the bowl, smooth object transfer, and collision-free approach and retreat trajectories in close proximity to the support arm.

**(2) Mix-with-Whisk:** The right arm retrieves a whisk from a nearby basket, mixes the contents of the bowl, and returns the whisk to its original location. This task involves tool use, curved and sustained motion, and close-proximity coordination with the left arm, which dynamically maintains the bowl pose throughout the sequence.

These tasks pose significant challenges for real world bimanual coordination. Both arms must operate in close proximity, requiring precise spatial alignment to avoid collisions–especially during approach and retreat phases. With only a single fixed camera and no wrist-mounted sensors, the policy must infer depth and object interactions purely from visual input. Timing mismatches or calibration drift between the arms can further compound errors, making successful execution sensitive to both perception and control stability.

Figure 11: **ViPRA-FM rollouts in real world bimanual tasks.** Top: `Place-in-Bowl` - the robot picks up a tomato and places it into a bowl held by the left arm. Bottom: `Mix-with-Whisk` - the robot retrieves a whisk, stirs the bowl contents, and returns the tool. Each sequence shows 10 evenly spaced frames sampled from real world executions.

## H.2    BIMANUAL RESULTS

ViPRA-FM is deployed using 14-step action chunks, executed at 7Hz control frequency. This high-frequency chunked control allows the policy to maintain smooth, temporally coherent trajectories while remaining responsive to changing visual inputs. The model also receives short history windows as input, which helps stabilize motion during contact-heavy transitions and multi-step interactions.

In `Place-in-Bowl`, the robot completes 10 out of 18 trials. Failures were primarily due to unsuccessful grasps caused by the limited span and compliance of our custom 3D-printed gripper, not the bimanual coordination itself. In all successful grasps, the object was consistently placed into the bowl without collision or instability. This suggests that the policy reliably handles the spatial reasoning and coordination demands of the task, with grasp robustness being the primary bottleneck, a limitation that could be mitigated with a more capable gripper design.

In `Mix-with-Whisk`, the robot completes 8 out of 12 trials. The task involves sustained, curved motion in close proximity to the left arm, requiring continuous spatial alignment between the whisk and bowl. The policy leverages temporal history to stay anchored to the mixing target and uses its chunked control output to produce smooth stirring behavior. The whisk's small, symmetric handle makes it easier to grasp, allowing the policy to focus on trajectory accuracy and contact stability throughout the sequence.

Together, these results demonstrate that ViPRA-FM is capable of executing complex bimanual tasks using a single vision-conditioned policy and continuous action generation. Additional results, comparisons, and rollout videos will be shared on our project website. `https://vipra-project.github.io`.

