# OpenReview forum: "ViPRA: Video Prediction for Robot Actions"
_ICLR.cc/2026/Conference — ICLR 2026 Poster_

### Official Review · Reviewer_1YWh · 2025-10-27

**Soundness:** 3
**Presentation:** 3
**Contribution:** 3
**Rating:** 6
**Confidence:** 3

**Summary:**

This paper introduces ViPRA, a hierarchical video-based robot control framework that learns discrete latent action representations from large-scale passive human and robot videos and then maps these to continuous robot actions via a flow-matching decoder. The approach jointly predicts future visual observations and latent action sequences during pretraining, and requires only a small amount of teleoperated data for downstream adaptation. The method is evaluated on SIMPLER simulation tasks and several real-world tabletop manipulation tasks, showing improved performance over recent latent-action and vision-language-action baselines.

**Strengths:**

- The work provides evidence that hierarchical visuomotor control can benefit from large-scale passive human and robot videos, reducing dependence on action-labeled demonstrations.

- The system attains competitive performance on physical manipulation tasks with only 100–200 teleoperated demonstrations, suggesting good data efficiency in downstream adaptation.

- The integration of flow-matching for action decoding leads to stable, high-frequency command generation suitable for real hardware, addressing common control smoothness limitations in discrete latent policies.

- The paper includes both simulation (SIMPLER) and real-world evaluations, with comparisons against several strong recent methods in the latent-action and VLA literature.

**Weaknesses:**

- **Limited technical novelty.**
Aside from data sacle, this work appears to be primarily an engineering integration of existing components rather than a new conceptual contribution (i.e. latent action learning, chunked action decoding, and flow-matching control) Similar hierarchical designs have been explored in recent systems such as UniVLA, LAPA, and UniPI.

- **Scaling claim not sufficiently validated.**
The paper attributes improvements to pretraining with large-scale passive human videos, but no controlled comparison is provided against training with robot-only video or reduced data subsets. Such experiments would help confirm that data scale, rather than architecture or training strategy, drives the gains.

- **Generalization scope is limited.**
The approach is framed as cross-embodiment and generalist, yet evaluations focus on similar 7-DoF manipulators and relatively simple tabletop tasks (pick and place). Broader embodiment diversity or clearer claim boundaries would improve alignment between motivation and evidence.

- **Ablation coverage.**
The system includes multiple engineered components (optical flow consistency, VQ bottleneck, multi-stage training). More detailed ablations isolating the effect of each would aid in understanding which design elements are critical to performance.

- **Compute and practicality not discussed.**
Although labeled data requirements are low, the approach requires substantial large-scale video pretraining. Reporting approximate compute/time requirements would help practitioners assess feasibility and compare with alternative methods.

Overall, this work offers a valuable scaling demonstration with solid empirical results, but the methodological novelty is limited, placing it at a borderline accept.

**Questions:**

- One of the most exciting implications of this work is that dynamics priors learned from human interactions may transfer across embodiments. Do the authors have any preliminary evidence or insights into what kinds of human video content (e.g., fine manipulation vs. gross motion) most help downstream control? Any failure cases that suggest limits of human-to-robot transfer?

- Have the authors attempted to manipulate latent action tokens intentionally to achieve specific motion patterns (e.g., alter motion direction, smoothness, or speed)? Observing consistency or interpretability here could reveal whether the learned latents are physically meaningful beyond reconstruction.

- Since the method leans heavily on scaling with passive data, at what point does additional video stop helping? Are there any observed regressions or negative-transfer effects when including noisier human videos from less structured environments?

---

> ### Author Response · Authors · 2025-11-22
> **Official Comment by Authors Part 1**
>
> ## Addressing technical novelty concerns
>
> | Component / Results     | **ViPRA**  |    **UniVLA**    |  **LAPA** |
> | ------------ | :-------:  | :-----: | :---: |
> | Fine primitive motion oriented latents      |     ✓     |  ✗ (temporally coarse) | ✗ (temporally coarse) |
> | Optical-flow consistency loss to pick up motion cues    |     ✓     |        ✗     |  ✗ |
> | Video-language backbone with native dynamics priors (LWM)  |     ✓     |   ✗ (Prismatic VLM, no video prediction)   | ✗ |
> | Multi-step latent action prediction during pretrain    |     ✓     | ✗  |  ✗ |
> | Joint future-frame latent prediction during pretrain      |     ✓     |             ✗             |  ✗  |
> | Flow-matching decoder for smooth continuous control        |     ✓     |         ✗ (L1 decoder)    |  ✗ (Discrete AR)  |
> | Avg. success rate on SIMPLER      |     69.8\% (AR) / 62.5\% (FM)   |         42.7\%    |  53.1\%
>
> | **Component / Results**      | **ViPRA** | **UniPI** |
> |--------|:----:|:----:|
> | Motion-centric latent actions    | ✓         | ✗         |
> | Future State/Video predictions     | ✓         | ✓         |
> | Uses Actionless/Human Videos    | ✓         | ✓         |
> | Supports high control frequency      | ✓         | ✗         |
> | Extensive real world results      | ✓         | O (very slow) |
> |  Avg. success rate on SIMPLER     | 69.8\% (AR) / 62.5\% (FM) | 1.7\%
>
> We respectfully disagree with the assessment that ViPRA is a hybrid of existing works like LAPA \[1\], UniVLA\[2\] and UniPI \[3\]. We summarize the core differences in the table above. Taken together, these form a substantial set of changes that makes ViPRA stand on its own as a distinct and meaningful method beyond prior work. We provide detailed comparison with each method below
>
> ### Comparision with LAPA and UniVLA
>
> - **Reformulating latent actions for fine-grained dynamics**: LAPA's \[1\] latent actions are **temporally coarse**—for instance, when using SSv2, they train with a 30-frame skip (\~2.5s). This aligns with their use of LWM \[4\] in Stage 2, where the objective is to predict $p(z\_t \\mid o\_t, c)$, treating latent action prediction as a **static image understanding** task. Similarly, UniVLA \[2\] also learns coarse latent actions from pairs of frames $(o\_t, o\_{t+k})$ spaced \~1 second apart, and train in a frozen high-level **DINOv2 feature space**. Such long gaps emphasize coarse, task-level changes but smooth out fine-grained motions.
>
>     However, LWM \[3\] is also capable of **video understanding and prediction**. To leverage this, we reformulate the latent action space in ViPRA to capture **fine-grained temporal dynamics** (at 4 to 6 Hz) over short a sequence of frames. This change is crucial for modeling high-frequency motions and provides robust and smooth priors for downstream policy.
>
> * **Unified training across embodiments**: Unlike LAPA \[1\], which trains separate latent action models for human and robot videos, ViPRA learns a **shared latent action space** across both domains. This unification promotes **cross-embodiment generalization** and enables seamless human-to-robot transfer. We make this concrete with the following evidence on SIMPLER
>
>    | Method | Pretraining data | Success Rate
>    |---|---|---
>    | LAPA | Human Videos (SSv2) | 52.1 \[1\]
>    | LAPA | Robot Videos (OpenX) | 53.1 (Evaluated by us)
>    |ViPRA-AR | Human + Robot Videos | **69.8**
>    |ViPRA-FM | Human + Robot Videos | **62.5**
>
> * **Action chunking**: Unlike LAPA \[1\] and UniVLA \[2\], ViPRA introduces chunk-level prediction for both latent action pretraining and real-action fine-tuning. Our ablations (Table 2) show that chunked latent modeling significantly improves downstream control: *ViPRA -AC* (59.2\%) -> *ViPRA* (69.8\%).
>
> * **Robust high-frequency control**: In ViPRA, we aimed to build a robust policy capable of handling **high-frequency, dexterous tasks**. As shown in Section 5.5 (Figure 6), ViPRA's continuous actions, learned via flow matching, are significantly more smooth and stable under noise compared to LAPA's discrete actions, which are brittle to visual perturbations. Moreover, LAPA \[1\] operatse at a inference frequency of less than 4.0 Hz as it performs single-step action prediction and they are not amenable to chunked action decoding. Similarly, UniVLA \[2\] action decoder gets capped at 7Hz. In contrast, ViPRA's flow matching decoder, with intelligent KV caching, can do inference at 1.95 Hz for the *entire action chunk*, enabling control frequency upto **22 Hz** (chunk size 14) on robot.

---

> ### Author Response · Authors · 2025-11-22
> **Official Comment by Authors Part 2**
>
> ### Comparision with UniPI
>
> UniPI \[3\] couples image and action prediction through a diffusion pipeline: it first generates future video with a diffusion model, then applies an inverse dynamics model (IDM) to recover actions. This design leads to low inference frequency, as documented in Appendix G.5. ViPRA targets high-frequency continuous control, which is not the focus of UniPI's architecture, but we include UniPI for completeness as a video-prediction baseline. In addition, UniPI's video prediction is often brittle on multi-step manipulation sequences, making it unsuitable for long-horizon control. This is reflected in its poor SIMPLER performance of 1.7%, which aligns with previous observations that diffusion-based video rollouts can accumulate errors quickly under embodied settings.
>
>
> ## Substantiating generalization claims
>
> We would like to highlight that our experiments span multiple embodiments:
> - A 6 DoF WidowX arm used in SIMPLER benchmarks.
> - A 7 DoF Franka (real world) single-arm manipulation tasks.
> - A bimanual Franka 14 DoF setups described in Appendix H, involving coordinated dual-arm skills such as transferring an object between two arms (Place-in-Bowl task), and performing sustained motion with arm coordination (Mix-with-Whisk task).
>
> Importantly, the pretraining data contained no bimanual demonstrations, only single-arm and human-video clips, yet ViPRA successfully transferred to these 14 DoF settings with minimal adaptation. This demonstrates that the learned motion-centric latent actions generalize beyond embodiment-specific kinematics and capture transferable priors. We will (i) summarize these results in the main text, and (ii) clarify in Section 5.4 that the bimanual evaluation highlights genuine embodiment transfer rather than domain-specific finetuning.
>
> Conceptually, ViPRA achieves embodiment generalization through motion-centric latent actions, providing a unified and physically grounded representation space that encodes local dynamics shared across embodiments, tasks and settings. Within short temporal windows, with frames sampled at 3 to 6 Hz, motion primitives such as moving left/right or reaching toward an object are similar across arms, hands, or robots. By formulating pretraining as learning a policy in this shared latent space, ViPRA enables transfer of motion knowledge from heterogeneous human and robot datasets into a single model that can quickly adapt to new embodiments and control interfaces.
>
> Moreover, ViPRA exhibits zero-shot generalization to novel objects not seen during pretraining or teleoperation, and shows emergent recovery behavior during long-horizon manipulation. For instance, on the Cover-Object task, the policy repeatedly retries until it successfully grasps the cloth, achieving a 100% pickup rate on this stage of the task. ViPRA also adapts to external perturbations applied mid-rollout, recovering the trajectory without manual resets. These examples provide evidence of emergent generalization axes beyond the nominal training settings. We provide additional examples and discussion in Appendix G, along with videos on the project website.
>
>
> ## Clarifying compute time
>
> We provide full compute and wall-clock requirements in Appendix B (Table 3) and Appendix C (Table 4). The latent action learning stage requires **168 hours on 8×H100 GPUs**. The video-language pretraining stage (7B backbone, latent-action + future-state prediction) requires **144 hours on 8×H100 GPUs** with a global batch size of 512. These costs are substantially lower than systems trained purely on action-labeled data. For context, OpenVLA \[5\] reports **21,500 A100-hours** for pretraining with a batch size of 2048.
>
>
> ViPRA’s efficiency stems from the structure of its training objectives. First, the future-state prediction loss leverages the model’s existing video-prediction capability and provides a dense supervision signal that accelerates learning of dynamics-aware representations. Second, the latent action bottleneck introduces a compact, motion-centric space (on the order of $8\^{16}$) that is substantially easier to model than the large discrete action vocabularies used in action-labeled policies (for example, $256\^{8}$). Together, these design choices enable ViPRA to learn a strong visuomotor prior from passive video with far fewer optimization steps and lower overall compute.

---

> ### Author Response · Authors · 2025-11-22
> **Official Comment by Authors Part 3**
>
> ## Exploring scaling claims
>
> We agree that understanding the relative contribution of human versus robot videos is important for characterizing ViPRA's generalization behavior. To study this, we are running two additional latent action pretraining runs: one using only human videos from SSv2 and one using only robot videos, both with identical architecture and training settings as our main model. These ablations directly isolate how each source contributes to latent space organization and downstream control. They are still in progress due to the high training cost for each run, but we will report preliminary numbers in the updated rebuttal and include full results on SIMPLER in the camera-ready version. Our expectation, based on the SIMPLER results where ViPRA already exceeds LAPA's human-only and robot-only models despite using fewer robot trajectories, is that both sources contribute complementary motion cues, and quantifying this explicitly is part of our planned analysis.
>
> ## Design ablations
>
> We provide comprehensive ablations in Table 2 (Section 5.5) covering the major design choices in ViPRA: action chunking, future state prediction, latent action prediction and the interaction between the two. These experiments isolate the effect of each component and make clear which elements are critical for downstream policy performance.
> Key takeaways are summarized below:
> * The LAPA baseline (latent-only) achieves 53.1%, whereas adding future state prediction in *ViPRA –AC* boosts performance to 59.2%, showing that **future state prediction provides a clear improvement** even with 1-step latent actions.
> * Removing state prediction from the full ViPRA setup (*ViPRA –SP2*) also leads to a 10.4% drop (69.8% → 59.4%), further confirming the importance of state prediction for effective policy transfer.
> * With the only state prediction variant *ViPRA –LA* (no latent action loss), we obtain 60.7 \% which is similar to *ViPRA -AC*, but still lower than full ViPRA.
> * Conversely, adding action chunking but removing state prediction (*ViPRA -SP2*, predicting 14-step action during pretraining and 14-step actual actions during fine-tuning) achieves 59.4%.
> * Our full ViPRA framework, combining both latent action chunking and state prediction, reaches 69.8%, clearly demonstrating that **latent action chunking and state prediction complement each other**.

---

> ### Author Response · Authors · 2025-11-22
> **Official Comment by Authors Part 4**
>
> ## Latent action ablations
>
> Regarding latent action learning stage, we also investigated the effect of optical flow consistency losses.
>
> | Setting             | Perplexity ↓   | Action MSE ↓  |
> | ------------------- | ------------  | ------------- |
> | Default (with flow) | **5.01**      |   **0.84**    |
> | No flow loss        | 5.63          |   0.92        |
>
> **What perplexity measures.**
> Perplexity is defined as $2^{H(p_i(k))}$, where $H(p_i(k))$ is the entropy of distribution over codebook indices at position $i$ (out of $N_{latent}$ positions). Its minimum is 1 (only one code used) and its maximum is $K$ (all codes used uniformly). However, for motion-centric latent spaces, high perplexity is not always desirable: it may reflect that static background information is leaking into the codes, forcing the model to represent appearance in codebooks rather than reserving capacity for dynamics.
>
> **What Action MSE measures.**
> To assess whether the learned latent actions contain information relevant for downstream robot control, we train a probe that maps the frozen codebook embeddings to normalized ground-truth robot actions for BridgeV2 dataset. This probe is trained after LAQ is fully trained and does not backpropagate into the encoder or codebook. Since LAQ never sees any action supervision during training, any predictive power in this probe reflects information that the latent space has implicitly organized through motion-centric self-supervision.
>
> **Role of perceptual loss.**
> Perceptual loss also plays a complementary role here. While we did not explicitly ablate it, intuitively it can be understood as capturing high-level appearance similarity. This makes it easier for the decoder $F\_\\alpha(\\hat{o}\_{t+1} | o\_{0:t}, z\_{0:t})$, which already has the input observations as conditioning, to reconstruct static background regions (e.g., walls, tables) without requiring detailed supervision from the latent space. This ensures that the capacity of the latent space would rather be utilized to capture *motion dynamics*.
>
> In our training pipeline, perceptual loss is introduced from the beginning to accelerate convergence of appearance reconstruction. Once good quality reconstructions are achieved, the flow loss is added to shift the model's focus toward encoding meaningful motion in the latent space. This staged training strategy ensures that codebook capacity is ultimately directed toward encoding dynamic cues essential for downstream policy learning.
>
> These results show that optical-flow supervision produces a more coherent and action-relevant latent space. With flow loss, the latent codes exhibit lower perplexity (5.01 vs. 5.63), indicating more consistent usage of the codebook across positions rather than diffuse or noisy assignments. More importantly, the action-probing experiment shows better alignment with ground-truth robot actions (MSE 0.84 vs. 0.92), even though the latent model never sees action labels. Together, these metrics suggest that flow guidance encourages the model to organize its latent capacity around motion dynamics rather than appearance, resulting in latent actions that are more predictive for downstream control.

---

> ### Author Response · Authors · 2025-11-22
> **Official Comment by Authors Part 5**
>
> ## Latent action analysis
>
> ### Evidence of latent action transfer and controllability
>
> Figure 5 provides direct evidence that ViPRA's latent actions encode motion dynamics. When latent actions extracted from an *UP*  video are injected into the forward deocoder model using a *DOWN* video starting frame, the model synthesizes a novel upward motion sequence. This cross-video transfer demonstrates that the latent actions modulate motion direction in a consistent and interpretable manner. Although we do not explicitly train for controllable editing, these rollouts show that the latent space captures motion structure suitable for downstream control. Figure 4 sheds further light on the structure of ViPRA’s latent action space. The positional codebook-usage maps show that specific tokens activate systematically across different motion directions.
>
> We provide further examples of such rollouts in the project website. There, we illustrate a similar effect appearing in our *LEFT* vs. *RIGHT* examples: injecting latent actions from a right-moving sequence into a left-moving video produces a novel right-moving rollout.
>
>
> ### Insight into cross-emodiment transfer
>
> ViPRA's cross-embodiment transfer arises from learning motion-centric latent actions over short temporal windows (3–6 Hz), which provide richer cues about underlying dynamics than two-frame latent approaches such as LAPA \[1\] and UniVLA \[2\]. Because the inverse model processes full clips rather than isolated frame pairs, the latents implicitly encode motion phase, approach direction, and interaction timing—signals that are consistent across hands, arms, and various robot manipulators. This helps explain why cross-embodiment pretraining benefits downstream robot performance: gross and mid-level motions such as reaching, lifting, and repositioning follow similar physical patterns across embodiments, and ViPRA leverages these shared dynamics more effectively than methods trained only on sparse frame differences. While very fine, contact-dependent adjustments are more embodiment-specific and ViPRA reconstructs these regions less accurately, the multi-frame design still preserves high-level context and motion cues that two-frame methods cannot capture, resulting in stronger overall transfer.
>
> ---
>
> \[1\] Ye et al., Latent Action Pretraining from Videos, ICLR 2025
>
> \[2\] Bu et al., UniVLA: Learning to Act Anywhere with Task-centric Latent Actions, RSS 2025
>
> \[3\] Du et al., Learning Universal Policies via Text-Guided Video Generation, NeurIPS 2023
>
> \[4\] Liu et al., World Model on Million-Length Video And Language With Blockwise RingAttention, ICLR 2025
>
> \[5\] Kim et al., OpenVLA: An Open-Source Vision-Language-Action Model, 2024

---

> ### Author Response · Authors · 2025-12-03
> **Official Comment by Authors Part 6**
>
> ## RE: Exploring scaling claims
>
> As outlined earlier, we began running additional latent-action pretraining ablations to isolate the contribution of human versus robot videos. These runs are now complete, and we report the preliminary results below. In all cases, both *latent action learning* and *pretraining* were performed using either (i) human-only videos or (ii) robot-only videos, with identical architecture and training settings as our main co-trained model.
>
> ### Ablation Results
>
> | Setting                  | LIBERO-10 ↑ | Action-Probe MSE ↓ |
> | ------------------------ | ----------- | ------------------ |
> | Co-train (Human + Robot) | **0.79**    | **0.84**           |
> | Robot-only               | 0.72        | 0.91               |
> | Human-only               | 0.69        | 0.99               |
>
> These ablations lead to three observations relevant to embodiment transfer:
>
> 1. **Human-only still transfers positively** (0.69 LIBERO). Performance is lower than robot-only, but still competitive, indicating that the short-horizon motion cues learned from human videos remain useful for robot control.
>
> 2. **Robot-only provides clean, embodiment-aligned signals**, leading to strong downstream performance and lower action-probe error. This aligns with expectations that robot trajectories map more directly to robot end-effector motion.
>
> 3. **Co-training achieves the best overall results**, suggesting that human and robot data offer complementary benefits: robot data grounds the latent space in robot kinematics, human videos introduce broader motion diversity, and, together they yield a richer motion-centric latent space than either source alone.
>
> ### Conclusion
>
> Overall, this supports the central claim that scaling passive video is a key driver of ViPRA’s improvements. ViPRA’s latent actions capture temporally fine-grained **motion primitives** that are consistent across embodiments. Human data, when combined with robot data, enriches the learned motion distribution and improves downstream control.

---

### Official Review · Reviewer_cNx8 · 2025-10-31

**Soundness:** 2
**Presentation:** 4
**Contribution:** 3
**Rating:** 4
**Confidence:** 3

**Summary:**

This work proposes a pretraining-finetuning framework, ViPRA, that learns discrete latent actions from videos without action labels by minimising perceptual and optical flow consistency objectives. Furthermore, this work proposes to use a flow matching decoder to convert the discrete actions to smoother continuous actions, which greatly prevent local discontinuities of previous works. The authors experiments on both simulation and real-world manipulation tasks and shown that the proposed method can achieve stronger performance.

**Strengths:**

- Very comprehensive related works, greatly aiding the readers in positioning the paper's contributions.
- The writing was easy to follow and clear, and while the background section is in the appendix, the context provided is enough to have a intuitive understanding of the proposed method.
- The proposed method regarding representation learning that leverages joint-prediction of pixels and latent actions seems intuitive and shows good performance.
- The proposed method regarding converting discrete actions to continuous control via flow matching is to my knowledge novel.
- Conducted comprehensive experiments and analysis, and the experiment results seem impressive (if we can ignore a major weakness for the moment).

**Weaknesses:**

- Major weakness: For Table 1 and Figure 3, which are the main results in the main manuscript, it is unclear how many seeds/runs/rollouts are experimented and how these values are calculated, so it is not possible to draw any conclusion about the (statistical) significance of the results presented at the moment.
- "Up to 22Hz" in the abstract and introduction is a bit misleading in my opinion, because the real-world experiments were done in 3.5Hz. (i.e. experiments with 22Hz were not done to show the efficacy).
- Since this work listed on real-world speed as a contribution, several comparison works (LAPA[1]) are not experimented on real world evaluations (i.e. Figure 3) but only in simulation, so it’s unclear how much speed is gained compared to previous works.
- Minor issues:
  - Figure 5 is introduced in the manuscript before Figure 4.
  - Table 6 is not referred nor explained (I assume it’s for appendix E.3, E.4). Furthermore, in Table 6 ViPRA is labelled but should be ViPRA-FM I think.
  - Apart from KV caching, which is more of an engineering technique for me, it is not obvious why inference speed is gained, since at inference the proposed method performs an extra flow-matching step compared to previous works.
  - The robot for the real world experiments in Figure 6/8 is not clearly written (I assume it’s Franka).
  - Not sure how to interpret the results where in LIBERO tasks, $\pi_0$ and UniVLA both outperform the proposed method ViPRA.

**Questions:**

I will repost some questions that were mentioned in the weakness section here for clarity.


$$\textbf{Suggestions}$$
S1: Clarify about how the evaluation metrics is calculated in Table 1 and Figure 3.
S2: It would be great if the authors can either amend the claim of 22Hz a bit, or perform some real world experiments at 22Hz.
S3: Could provide some clarity about ViPRA-AR since it's not really explained anywhere.
S4: Minor: I assume the optical flow model RAFT is pretrained and frozen although it is not explained anywhere. It would be great if the authors could clarify.
S5: Minor: Could improve page 22’s readability a bit by aligning them properly (e.g. at the top).

$$\textbf{Questions}$$
Q1: Some related works suggest that pixel-level reconstruction is somewhat not as efficient [1][2], what do the authors think about this? Since this work proposes to learn latent actions, can we argue that it would be more effective to learn latent representation features rather than the full pixel reconstruction?
Q2: If I understand correctly, in Table 1, compared to LAPA[3] and UniVLA[4], the proposed ViPRA performs quite better in full success rate, showing that leveraging video prediction is quite helpful to learn robust representations for SIMPLER tasks. However, in LIBERO’s experiments, UniVLA performs best. What do the authors think about this?
Q3: Following up on Q2, UniVLA is the best performing framework in LIBERO-10, the authors attributed this to UniVLA being optimised for LIBERO. Can the authors further clarify what this means? For example, what kind of performance can we expect on SIMPLER if we somehow optimise UniVLA for it?
Q4. Can we also use KV caching for other related works (LAPA, UniVLA) to get improved speed as well (I believe UniVLA already uses action chunking)? If so, it’s unclear to me how much speed is gained compared to previous works.

---

[1] G. Zhou et al., DINO-WM: World Models on Pre-trained Visual Features enable Zero-shot Planning, arXiv 2024
[2] R. Sun et al., Learning Latent Dynamic Robust Representations for World Models, ICML 2024
[3] S. Ye et al., Latent Action Pretraining from Videos, ICLR 2025
[4] Q.Bu et al., UniVLA: Learning to Act Anywhere with Task-centric Latent Actions, RSS 2025

---

> ### Author Response · Authors · 2025-11-22
> **Official Comment by Authors Part 1**
>
> We thank the reviewer for their assessment. Below we provide detailed clarifications for the questions that were raised in the review.
>
> ## Clarification on evaluation metrics
>
> **SIMPLER results in Table 1**: For all SIMPLER experiments, we evaluate 25 rollouts for each of the four tasks under three different random seeds, and we report the average success rate and grasp rate across these runs. We evaluate the discrete baselines Scratch-AR, OpenVLA, LAPA, and ViPRA-AR, as well as the continuous baselines Scratch-FM, $\pi_0$, ViPRA-FM, under this same protocol. For VPT and UniPI, we directly use the results reported in LAPA; for UniVLA, we use the numbers from their paper, which follow an equivalent SIMPLER evaluation setup.
>
> **LIBERO-10 results in Table 6**: Each of the ten tasks is evaluated over 10 episodes per task using 5 different random seeds, and we report the average reward aggregated over all 10 tasks (500 episodes total). The reviewer is correct that the row should be labeled ViPRA-FM rather than ViPRA, and we will correct this in the revision. For comparison methods, we use the results reported in their respective papers. Additionally, we evaluated UniVLA's publicly released LIBERO checkpoint under the same evaluation setup, and observed a success rate of 0.86, which is slightly lower than the 0.92 reported in their paper.
>
> **Real-world results in Figure 3.**
> We use two 7-DoF Franka Panda robots for all real world experiments. This is specified in Section 5.1 under "Real World Manipulation" with more details in Appendix G. Different tasks use different numbers of trials, and Appendix G provides additional details about tasks. For completeness, we list the exact number of evaluation episodes used for each task:
> * **Cover-Obj:**
>   We evaluate 18 distinct object configurations, varying object identity and placement order (including both seen and unseen objects).
>   Each configuration is attempted 3 times, for a total of **54 trials**.
> * **Pick-Place:**
>   We evaluate 6 distinct object configurations, again varying object identity and placement order.
>   Four different destination locations are used.
>   Each configuration is repeated 2 times, yielding a total of **48 trials**.
> * **Stack-Cup:**
>   This task exhibits the most diversity: we vary the initial and final cup positions as well as cup colors, following the task instructions.
>   We evaluate 64 unique combinations, each executed once, for **64 total trials**.
>
> We will incorporate these counts directly in Appendix G to avoid any ambiguity.
>
> ## Clarification on control frequency
>
> ### Justifying "upto 22Hz" claims
>
> We thank the reviewer for raising this point. The distinction between policy inference frequency, robot control frequency, and action chunking is important, and we clarify it here.
>
> In ViPRA, we aimed to build a **robust policy** capable of handling **high-frequency, dexterous tasks**. ViPRA-FM flow matching decoder predicts a chunk of 14 future continuous actions. On a single H100, the entire chunk is generated in 0.51 s, corresponding to an inference rate of 1.95 Hz for the whole chunk. Since each chunk contains 14 actions, this corresponds to a *potential robot control frequency* of 14 × 1.95 Hz ≈ 27.3 Hz if actions are executed sequentially on the robot without additional latency. We also have to consider image processing times, converting images into discrete tokens through LWM's \[1\] VQGAN tokenizer, and language tokenizations times, leading to the conservative estimate of "upto 22Hz". This analysis is similar to the one reported in OpenVLA-OFT \[2\] (Table III).
>
> Our actual hardware runs, other delays also arise from: (i) server–robot communication delays, (ii) image transmission time, and (iii) the Franka control stack, not by the ViPRA policy itself, further limiting the end to end throughput.

---

> ### Author Response · Authors · 2025-11-22
> **Official Comment by Authors Part 2**
>
> ### Why ViPRA achieves higher effective speed than other latent action baselines
>
> **ViPRA vs LAPA**:
> LAPA's discrete policy is fundamentally single-step and not amenable to chunking. LAPA produces discrete 7-DoF (or 14-DoF for bimanual) action by autoregressively generating one discrete token per dimension. **Even with KV caching**, this architecture requires one forward pass per action dimension, which caps throughput at roughly 4 Hz for a single action. The major bottleneck being they have to run one forward pass per action dimension.
>
> Extending this design to multi-step prediction is computationally prohibitive: producing a 14-step action sequence would require decoding
> 14 × 7 = 98 discrete tokens sequentially, pushing inference well below 1 Hz. This limitation is not incidental and stems directly from LAPA's pretraining objective, which infers one latent action from two frames without modeling short-horizon motion context. Consequently, the policy is optimized only for one-step prediction, and its discrete codebook is not structured to support multi-step generation.
>
> Therefore, LAPA cannot benefit from chunked decoding, and adding KV caching alone does not remove its fundamental architectural bottleneck.
>
> **ViPRA vs UniVLA**:
> Like LAPA, UniVLA’s latent action pretraining is strictly single-step, inferring one latent action from two frames, yet its finetuning modifies the task to predict 10-step action chunks in one forward pass. Moreover, they train their latent action model in frozen DINOv2 space, which smooths out fine-grained motions. This creates a pretraining–finetuning mismatch, since the latent space was never optimized to represent or propagate short-horizon motion, even though it is later used for multi-step control. In our setup, this forced chunking enables UniVLA to reach about 7 Hz, but the multi-step behavior is not grounded in a learned latent dynamics model. By contrast, ViPRA’s latent actions are learned via perceptual and optical-flow consistency on video clips, explicitly capturing short-horizon temporal structure, so ViPRA-FM’s chunked rollout is directly aligned with its pretraining and leverages motion-centric latent cues rather than relying on post-hoc chunking.
>
> ### Why real-world evaluations were conducted at 3.5 Hz
>
> For real world experiments, we initially selected a control rate of 3.5 Hz to ensure a fair and safe comparison with discrete autoregressive baselines such as OpenVLA and LAPA. These baselines inherently operate below 4 Hz, since they decode actions one dimension (or one step) at a time and cannot leverage chunked or multi-step inference. Running ViPRA at a substantially higher frequency would therefore make the comparison unfair, as these baselines cannot be deployed at such rates on hardware.
>
> However, as detailed in the paper, discrete AR policies also exhibit abrupt action jumps and instability under visual noise, as shown in Section 5.5, Figure 6, with even more details in Appendix F. In our real world setup, these issues frequently triggered the Franka robot's safety stops, making evaluation of discrete baselines unsafe.
>
> To further illustrate ViPRA-FM's capability, we provide two additional real world rollouts of the same task side-by-side on the project website: one executed at 3.5 Hz and another at 7 Hz. ViPRA-FM seamlessly supports the higher control rate, completing the task 38\% faster with visibly lower inference latency. These demonstrations highlight that ViPRA-FM can operate at high frequencies in practice, even though our official evaluations are kept at 3.5 Hz for fairness and safety.
>
> ### Why KV caching is not a simple engineering additions
>
> Flow matching policies such as $\pi_0$ typically follow a two system design: a VLM backbone computes visual-language embeddings, and a separate action expert model uses these embeddings to perform the flow matching iterations to get actions. This introduces architectural duplication and prevents fully unifying the model into a single transformer backbone.
>
> In ViPRA, we instead integrate the noise encoder and flow matching action decoder directly into the same LWM \[1\] transformer backbone. This avoids the two system setup entirely: both the perception context and the action generation utilize the same transformer backbone optimized by finetuning. This makes KV caching a natural substitute for the two system design, because a single transformer backbone handles both context encoding and action generation, and its cached key–value states are reused throughout the flow matching process.
>
> Thus, KV caching in ViPRA is not a simple engineering addition—it is made effective by the architectural choice to remove the two system design and unify perception and action inside the same transformer, simplifying the pipeline and enabling efficient multi-step action rollout.

---

> ### Author Response · Authors · 2025-11-22
> **Official Comment by Authors Part 3**
>
> ## Miscellaneous clarifications
>
> **RAFT for optical flow.**
> For the optical flow consistency loss, we use RAFT-Large, a widely adopted model pretrained on FlyingChairs/FlyingThings3D and finetuned on Sintel, KITTI, and HD1K, with competitive endpoint error (EPE) of 1.82 on Sintel-Clean and 3.07 on Sintel-Final. We use this model **fully frozen**. The flow loss is applied with gradient clipping and a small weighting factor, so occasional noisy estimates do not destabilize training.
>
> **UniVLA being optimized for LIBERO.**
> This is actually an oversight on our part and we thank the reviewer for pointing this out. We actually meant UVA here which reports 0.90 success rate on LIBERO-10. We have fixed this and also included UVA in Table 6 now. We would like to point out that UVA regenerates LIBERO actions in absolute end-effector (EEF) space by replaying LIBERO demonstrations inside the simulator, and its policy is trained directly in this absolute action space. In contrast, ViPRA operates in the standard delta EEF action space without proprioception or wrist-view inputs, which is a more challenging setting due to cumulative drift from purely image-based delta actions. This makes a strict one-to-one comparison difficult. Moreover, UVA conducts both video generation and joint video+action prediction in-domain on LIBERO, while our ViPRA checkpoint was pretrained on a broad collection of real human and robot videos and then directly finetuned on LIBERO without repeating latent action pretraining in that domain.
>
> **UniVLA outperforming ViPRA on LIBERO**: We agree with the reviewer that UniVLA achieves higher performance on LIBERO-10. We believe the difference in LIBERO-10 performance reflects differences in data scale and model complexity, not a limitation of ViPRA's latent action formulation.
> - UniVLA pretrains on a larger and diverse dataset combining the entire of OpenX Embodiment manipulation (vs ViPRA which uses only 3 datasets from OpenX), GNM \[3\] navigation, and Ego4D \[4\] egocentric videos. This gives UniVLA pretraining a strong advantage over ViPRA.
> - Flow matching is slower to train compared UniVLA simple L1 action decoder. It is also more sensitive to finetuning hyperparameters like batch size, iteration count, learning rate and scheduler choice. So far, our best run has attained 0.81, which is pretty close to the 0.86 we measured when evaluating UniVLA's publicly released under the same protocol.
> - Despite these constraints, ViPRA-FM achieves 0.62, outperforming UniVLA's reported 0.42 on SIMPLER considerably. This highlights that ViPRA's motion-centric latent actions offer robust priors for downstream control.
>
> **Clarifying ViPRA-AR**: As described in Section 5.5, ViPRA-AR mirrors the discrete-action formulation used in LAPA and OpenVLA, but extends it to support action chunking. It predicts a 14-step action chunk, with each action represented as 7 discrete tokens (7-DoF), yielding 98 tokens per chunk, making ViPRA-AR inherently slower to decode than the continuous ViPRA-FM. For this reason, we evaluate ViPRA-AR only in simulation. Despite its slower discrete decoder, ViPRA-AR performs competitively, showing that ViPRA’s motion-centric latent actions remain effective across both discrete and continuous action heads.
>
> **Comparison with non-pixel reconstruction works**: We thank the reviewer for this insightful question and for highlighting these relevant works \[5, 6\]. While we agree that pixel-level reconstruction can be computationally intensive compared to latent-only approaches, we found it essential for our specific goal of learning *abstract* robot actions from actionless videos. Unlike standard world model settings where explicit actions are available to shape the latent space, ViPRA operates in an unsupervised setting where visual evolution serves as the only ground truth signal. Pixel reconstruction provides a dense and self-supervised objective that forces our latent actions to capture all significant dynamics in the scene. If we only optimized for high-level latent features, such as the DINO embeddings used in \[5\] (and even in UniVLA), the model might ignore subtle but physically critical motion details. These details, such as precise contact dynamics or minor object shifts, are necessary for low-level control but are sometimes invisible to high-level semantic encoders.
>
> ---
>
> \[1\] Liu et al., World Model on Million-Length Video And Language With Blockwise RingAttention, ICLR 2025
>
> \[2\] Kim et al., Fine-Tuning Vision-Language-Action Models: Optimizing Speed and Success
>
> \[3\] Shah et al., GNM: A General Navigation Model to Drive Any Robot, ICRA 2023
>
> \[4\] Grauman et al., Ego4D: Around the World in 3,000 Hours of Egocentric Video
>
> \[5\] Zhou et al., DINO-WM: World Models on Pre-trained Visual Features enable Zero-shot Planning, arXiv 2024
>
> \[6\] Sun et al., Learning Latent Dynamic Robust Representations for World Models, ICML 2024

---

### Official Review · Reviewer_mq1Q · 2025-11-01

**Soundness:** 3
**Presentation:** 2
**Contribution:** 2
**Rating:** 4
**Confidence:** 3

**Summary:**

This paper proposes ViPRA, a framework for learning robot policy from action-free videos by incorporating video prediction and latent policy learning. The key idea is to pretrain a video–language model to jointly predict both (i) future visual frames and (ii) motion-centric latent actions that summarize local dynamics, guided by perceptual and optical flow consistency losses. These latent representations are mapped to continuous action space through a flow-matching decoder trained on a small number of teleoperated demonstrations.
The paper claims that this “video prediction + latent action” pretraining allows robots to leverage large-scale unlabeled human and robot videos, achieving improvements on both simulation benchmark and real-world tasks.

**Strengths:**

The paper tackles a major challenge in robot learning—leveraging large-scale actionless videos for control. Using video prediction to inject physical dynamics into latent actions is conceptually coherent and builds upon trends in world-model-based control. Evaluation spans both simulation (SIMPLER benchmark) and real-world manipulation (Franka bimanual setup). ViPRA gets better performance comparing against plausible baselines including LAPA, UniVLA, π0, and diffusion-policy variants.

**Weaknesses:**

1. **Contribution**
- Unclear novelty: It is not fully clear whether ViPRA introduces a fundamentally new paradigm, or whether it can be viewed as a hybrid of LAPA (latent action tokenization) and UVA (unified video and action prediction).

2. **Codebook Design**
- Codebook size (|C| = 8) appears extremely small compared to typical VQ-based latent action works (e.g., 128–8192).
- No ablation or justification is provided for this choice, nor evidence that such a small capacity suffices to capture motion diversity.
- The paper should include:
(i) Ablations over codebook size (8 / 32 / 128 / 512 / 8192).
(ii) Quantitative codebook utilization metrics (entropy, perplexity, diversity).


3. Data Scale, Composition, and Scalability
- Scaling behavior: Although the model claims to leverage “large-scale actionless videos,” the dataset (~400K clips) remains moderate.
There is no scaling analysis showing how performance varies with the number of pretraining videos.
- A performance–vs–data-size curve would clarify whether ViPRA is still data-limited.
- Human–robot ratio: The impact of mixing human and robot videos is not studied. Different ratios may have large effects on transfer and generalization; sensitivity curves or ablations are needed.
- Generalization limitation: Without scaling or compositional studies, it is unclear if the model can extend to larger internet-scale video corpora.

4. Latent–Action Semantics and Alignment
- The alignment between learned latent actions and ground-truth actions (on datasets where GT is available) is not quantified, which would verify whether latent tokens actually capture actionable dynamics rather than visual motion.
- Would the latent action learned from multiple sources enable a unified action space between multiple embodiments?

5. Optical-Flow Supervision and Robustness
- The $L_{flow}$  loss is claimed in the contribution but lacks ablation and robustness analysis.
- How much performance degrades without L_flow?
- How stable is RAFT-based flow under high ego-motion, blur, or occlusion?
- Would learned or multi-frame flow estimators perform better?

6. Missing or Incomplete Baselines
- UVA is cited but not directly compared, despite strong overlap in video-conditioned policy learning. Including UVA as an explicit baseline would help contextualize improvements.

**Questions:**

Please refer to the weakness part. I would like to raise my score once the concerns are resolved.

---

> ### Author Response · Authors · 2025-11-22
> **Official Comment by Authors Part 1**
>
> We thank the reviewer for their assessment. Below we provide detailed clarifications for the questions that were raised in the review.
>
> ## Novelty relative to LAPA and UVA
>
> | Aspect                                            | **LAPA**                            | **UVA**                       | **ViPRA (ours)**                                           |
> | ---------------------- | --------------------- | ------------------- | ----------------- |
> | Does not use action labels during pretraining             | ✓           | ×                | ✓       |
> | Uses large-scale actionless human videos effectively          | × (separate human/robot tokenizers) | ~ (~3k videos; used in simulation only; marginal gain) | ✓ (unified latent space across 400k human + robot videos)  |
> | Temporal granularity of latent actions            | × (coarse 30-frame skip)      | × (no latent actions)   | ✓ (fine-grained short-horizon latent actions at 3 to 6 Hz)                     |
> | Flow-based supervision for motion-centric latents | ×       | ×          | ✓      |
> | Joint video + latent action modeling              | ×                 | ~ (requires real actions)          | ✓ (without action labels)                                  |
> | Multi-step latent action prediction during pretraining              | ×        | ~ (requires real actions)      | ✓           |
> | High-frequency continuous robot control           | × (up to 4 Hz, discrete AR)              | ~ (multi-step DDPM; slower)   | ✓ (flow matching; up to 22 Hz)    |
> | Action chunking for smooth trajectories  | ×          | ✓           | ✓
> | Cross-embodiment generalization                   | ~ (separate human/robot tokenizers)             | × (real data on UMI only)      | ✓ (shared latent space across human + multiple robot arms + viewpoints) |
>
> Legend: ✓ (yes)  /  ~ (limited or partial)  /  × (no)
>
> We respectfully disagree with the assessment that ViPRA is a hybrid of LAPA \[1\] and UVA \[2\]. We summarize the core differences in the table above. Taken together, these form a substantial set of changes over LAPA \[1\] and UVA \[2\]. We believe ViPRA stands on its own as a distinct and meaningful method beyond prior works. We provide detailed comparison with each method below.
>
> ### ViPRA vs LAPA
>
> * **Reformulating latent actions for fine-grained dynamics**: LAPA's \[1\] latent actions are **temporally coarse**—for instance, when using SSv2, they train with a 30-frame skip (\~2.5s). This aligns with their use of LWM \[3\] in Stage 2, where the objective is to predict $p(z_t \mid o_t, c)$, treating latent action prediction as a **static image understanding** task. However, LWM \[3\] is also capable of **video understanding and prediction**. To leverage this, we reformulate the latent action space in ViPRA to capture **fine-grained temporal dynamics** (at 4 to 6 Hz) over short horizons. This change is crucial for modeling high-frequency motions and provides robust and smooth priors for downstream policy.
>
> * **Unified training across embodiments**: Unlike LAPA \[1\], which trains separate latent action models for human and robot videos, ViPRA learns a **shared latent action space** across both domains. This unification promotes **cross-embodiment generalization** and enables seamless human-to-robot transfer. We make this concrete with the following evidence on SIMPLER
>
>    | Method | Pretraining data | Success Rate
>    |---|---|---
>    | LAPA | Human Videos (SSv2) | 52.1 \[1\]
>    | LAPA | Robot Videos (OpenX) | 53.1 (Evaluated by us)
>    |ViPRA-AR | Human + Robot Videos | **69.8**
>    |ViPRA-FM | Human + Robot Videos | **62.5**
>
> * **Action chunking**: ViPRA introduces chunk-level prediction for both latent action pretraining and real-action fine-tuning. Our ablations (Table 2) show that chunked latent modeling significantly improves downstream control—*ViPRA -AC* (59.2\%) -> *ViPRA* (69.8\%).
>
> * **Robust high-frequency control**: In ViPRA, we aimed to build a **robust policy** capable of handling **high-frequency, dexterous tasks**. As shown in Section 5.5 (Figure 6), ViPRA's continuous actions, learned via flow matching, are significantly more smooth and stable under noise compared to LAPA's discrete actions, which are brittle to visual perturbations. Moreover, LAPA \[1\] operates at a fixed inference frequency of **4.0 Hz** (7 DoF), as it performs one forward pass per action dimension and is not amenable to chunked decoding. In contrast, ViPRA's flow-matching decoder, with intelligent KV caching, can do inference at **1.95 Hz** for the *entire chunk*, enabling control frequency upto **22 Hz** (chunk size 14) on robot.
>
>
> Notably, we also manage to address shortcomings of LAPA \[1\], acknowledged by its authors
> > *“LAPA underperforms compared to action pretraining when it comes to fine-grained motion generation tasks like grasping. … Second, similar to prior VLAs, LAPA also encounters latency challenges during real-time inference.”*
> >,  *Section 6.2, Limitations, LAPA \[1\]*

---

> ### Author Response · Authors · 2025-11-22
> **Official Comment by Authors Part 2**
>
> ### ViPRA vs UVA
>
> **Different motivation for future prediction**: While both ViPRA and UVA \[2\] have future prediction components, the design and purpose of that prediction differ substantially. When applied to action-free videos, UVA \[2\] essentially reduces to a **frame sequence completion task**, implicitly assuming that improved visual continuity will translate to better policies. They do not explicitly ground these visual changes in an action-centric representation. In contrast, ViPRA uses future prediction to explicitly encode **chunk-level motion dynamics**: we predict the **future state after a horizon** $o_{t+H}$ and then infer the sequence of latent actions $z_{t:t+H-1}$ that causally lead to it:
>   $$
>   p(o_{t+H} \mid o_t, o_{t-1}, c), \quad p(z_{t:t+H-1} \mid o_{t+H}, o_t, o_{t-1}, c)
>   $$
> This is consistent with our latent action learning objective in Stage 1, where latent actions are explicitly formulated to encode **fine-grained motion cues**. Such multi-step training provides robust motion-centric priors that transfer effectively to downstream policy learning.
>
> **UVA's limited data diversity**: UVA \[2\] only reports results on a narrow real world data distribution: **1,500 action-labeled demonstrations** from **three tasks**, all collected with a handheld UMI device—a highly **consistent visual setup**. Crucially, all data used includes action supervision. In contrast, ViPRA pretrains on 350k diverse, action-free videos, including 170k robot clips across three embodiments and viewpoints (Fractal, Kuka, Bridge) and 180k human videos (SSv2). Fine-tuning uses just 400 action-labeled demonstrations from three real-world Franka tasks.
>
> **Minimal action-free video usage in UVA**: Although UVA \[2\] supports action-free training in principle, it reports only limited use, *3,175 human videos* for pretraining on LIBERO-10; a simulation only benchmark. It only reports a **marginal improvement** in accuracy (0.90 -> 0.91, Table VIII). Thus, it provides limited evidence about efficiently using large-scale action-free videos for real world tasks.
>
>
> **Comparison with UVA**: We evaluated ViPRA on LIBERO-10 (results summarized in Appendix E), where ViPRA achieves a 0.79 success rate, outperforming OpenVLA and $\pi_0$-FAST and approaching UVA's \[2\] 0.90 despite using a substantially different supervision regime. We emphasize that UVA \[2\] regenerates LIBERO actions in absolute end-effector (EEF) space by replaying LIBERO demonstrations inside the simulator, and its policy is trained directly in this absolute action space. In contrast, ViPRA operates in the standard delta EEF action space without proprioception or wrist-view inputs, which is a more challenging setting due to cumulative drift from purely image-based delta actions. This makes a strict one-to-one comparison difficult. Moreover, UVA \[2\] conducts both video generation and joint video+action prediction in-domain on LIBERO, while our ViPRA checkpoint was pretrained on a broad collection of real human and robot videos and then directly finetuned on LIBERO without repeating latent action pretraining in that domain. Despite these differences, ViPRA remains competitive, and we expect the gap to close further by (i) finetuning with absolute LIBERO action labels and (ii) performing latent action + future state pretraining in domain. We will include these observations and UVA \[2\] numbers in the final revision.
>
> | **Method**       |  UVA | ViPRA |
> | ------------ | ----: | ------: |
> | **Success Rate** |   0.90 |  0.79 |

---

> ### Author Response · Authors · 2025-11-22
> **Official Comment by Authors Part 3**
>
> ### Codebook Size: Why Small is Intentional and Necessary
>
> We respectfully disagree with the reviewer's assessment that a codebook of size 8 is small or insufficient for latent action learning. In fact, all contemporary latent action works intentionally use very small codebooks for the same reason we do:
>
> ### Small codebooks are standard across all latent action literature
>
> | Method                                            | Purpose                                   | Codebook Size          |
> | ------------------------------------------------- | ----------------------------------------- | ---------------------- |
> | GENIE \[4\]  | latent world model / neural simulator   | 8                  |
> | LAPA \[1\]        | latent actions for robot control                       | 8                  |
> | UniVLA \[5\]    | latent-actions for policy learning          | 16                 |
> | AdaWorld \[6\]   | latent action for visual planning; continuous latent; small capacity    | 32 (single vector) |
> | ViPRA (ours)                                  | fine-grained short-horizon motion latents | 8                  |
>
> Across these works, and ours, the motivation remains same:
> *a small latent space (information bottleneck) is necessary to force the latent space to encode motion dynamics rather than appearance shortcuts.* A large codebook violates this bottleneck and destroys the abstraction.
>
>
> Finally, although the codebook itself is small, our latent representation uses 16 discrete indices ($N_{\text{latent}} = 16$), giving an effective latent space of $8^{16}$. As shown in Figure 4, the codebook entries are positionally sensitive, so capacity also comes from the structured multi-index representation rather than the size of the vocabulary itself.
>
>
> ### Why larger codebooks harm dynamics learning?
>
> In ViPRA, the latent encoder $I\_\\beta(z\_t \\mid o\_{0:L})$ observes a short history including future frames.
> If the codebook were large (128–8192 entries), the encoder could:
>
> * simply copy high-level appearance or future visual features into the latent token
> * memorize shortcuts that allow the decoder to reconstruct future states
> * avoid representing the underlying motion entirely
>
> This would make the latent space visually predictive, not **dynamically predictive**, which defeats the purpose of having latent actions. In our setting, the decoder $F\_\\alpha(\\hat{o}\_{t+1} \\mid o\_{0:t}, z\_{0:t})$ already receives the past observations as input. It does not need high-capacity latents to reconstruct the future. The latent should carry only motion information, not appearance. A larger codebook would create an information leak, leading to shortcut solutions where the latent encodes pixel-level details instead of dynamics.
>
>
> ### When large codebooks make sense?
>
> Large codebooks are appropriate for **image-level VQ models** (VQ-VAE \[7\], DALL-E \[8\], MAGVIT \[9\]), where the goal is to reconstruct high-dimensional appearance. But latent actions are not images. They are low-dimensional motion primitives, so, the vocabulary should be small, and targeted at dynamics, not texture or semantics.

---

> ### Author Response · Authors · 2025-11-22
> **Official Comment by Authors Part 4**
>
> ## Loss Analysis and action alignment
>
> We performed a targeted ablation at the latent action learning stage to investigate the effect of optical flow consistency losses.
>
> | Setting             | Perplexity ↓   | Entropy ↓ | Action MSE ↓  |
> | ------------------- | ------------  | ------------- | ------------- |
> | Default (with flow) | **5.01**      |  **1.59** | **0.84**    |
> | No flow loss        | 5.63          | 1.74  | 0.92        |
>
> **What perplexity measures.**
> Perplexity is defined as $2^{H(p_i(k))}$, where $H(p_i(k))$ is the entropy of distribution over codebook indices at position $i$ (out of $N_{latent}$ positions). Its minimum is 1 (only one code used) and its maximum is $K$ (all codes used uniformly). However, for motion-centric latent spaces, high perplexity is not always desirable: it may reflect that static background information is leaking into the codes, forcing the model to represent appearance in codebooks rather than reserving capacity for dynamics.
>
> **What Action MSE measures.**
> To assess whether the learned latent actions contain information relevant for downstream robot control, we train a probe that maps the frozen codebook embeddings to normalized ground-truth robot actions for BridgeV2 dataset. This probe is trained after LAQ is fully trained and does not backpropagate into the encoder or codebook. Since LAQ never sees any action supervision during training, any predictive power in this probe reflects information that the latent space has implicitly organized through motion-centric self-supervision.
>
> **Role of perceptual loss.**
> Perceptual loss also plays a complementary role here. While we did not explicitly ablate it, intuitively it can be understood as capturing high-level appearance similarity. This makes it easier for the decoder $F\_\\alpha(\\hat{o}\_{t+1} | o\_{0:t}, z\_{0:t})$, which already has the input observations as conditioning, to reconstruct static background regions (e.g., walls, tables) without requiring detailed supervision from the latent space. This ensures that the capacity of the latent space would rather be utilized to capture *motion dynamics*.
>
> In our training pipeline, perceptual loss is introduced from the beginning to accelerate convergence of appearance reconstruction. Once good quality reconstructions are achieved, the flow loss is added to shift the model's focus toward encoding meaningful motion in the latent space. This staged training strategy ensures that codebook capacity is ultimately directed toward encoding dynamic cues essential for downstream policy learning.
>
> These results show that optical-flow supervision produces a more coherent and action-relevant latent space. With flow loss, the latent codes exhibit lower perplexity (5.01 vs. 5.63), indicating more consistent usage of the codebook across positions rather than diffuse or noisy assignments. More importantly, the action-probing experiment shows better alignment with ground-truth robot actions (MSE 0.84 vs. 0.92), even though the latent model never sees action labels. Together, these metrics suggest that flow guidance encourages the model to organize its latent capacity around motion dynamics rather than appearance, resulting in latent actions that are more predictive for downstream control. The reviewer is right in suggesting that latent action learned from multiple sources enable a unified action space between multiple embodiments.
>
> **RAFT Stability and Alternatives.**
> For the optical flow consistency loss, we use RAFT-Large, a widely adopted model pretrained on FlyingChairs/FlyingThings3D and finetuned on Sintel, KITTI, and HD1K, with competitive endpoint error (EPE) of 1.82 on Sintel-Clean and 3.07 on Sintel-Final. We use this model fully frozen. In our data distribution (teleoperated robot videos and SSv2 human clips), camera motion is minimal and scenes are from third-person viewpoints, so RAFT provides stable flow in practice. The flow loss is applied with gradient clipping and a small weighting factor, so occasional noisy estimates do not destabilize training. Although we did not replace RAFT with multi-frame or learned flow models, we expect limited benefit in our setting because the datasets exhibit low ego-motion. We also acknowledge in Section 6 (Limitations and Future Work) that extending ViPRA to large-scale egocentric videos is an important direction; in that regime, more advanced multi-frame flow estimators could help capture ego-motion in the latent space and support short-horizon navigation or forecasting. Importantly, swapping RAFT for a stronger flow model would not change the ViPRA framework itself: the pretraining objective, latent action formulation, and downstream policy learning remain identical, with optical flow serving as a auxiliary motion cue.

---

> ### Author Response · Authors · 2025-11-22
> **Official Comment by Authors Part 5**
>
> ## Data scale and composition
>
>
> **Data scale.**
> Scaling ViPRA beyond 400k videos was not feasible under our compute and storage budget; as reported in Appendix B and C, both latent action learning and pretraining require over 7 days per run even at our current scale. Although we do not train   multiple models at different scales, our single ViPRA model already outperforms prior latent action baselines on SIMPLER despite using less robot data. In particular, ViPRA surpasses LAPA (human-only), LAPA (robot-only), and UniVLA, all of which are pretrained on substantially larger robot-trajectory datasets than ours. This suggests that ViPRA is able to extract strong motion-centric priors even within a moderate actionless video corpus. We agree that exploring larger pretraining corpora and studying explicit scaling curves is a worthwhile direction, and we highlight this in Section 6 (Limitations and Future Work). Our goal with ViPRA is to provide a practical blueprint for leveraging actionless videos for generalist robot control, and we expect future work using larger-scale datasets to further support these trends.
>
> **Data Compositiion.**
> We agree that understanding the relative contribution of human versus robot videos is important for characterizing ViPRA's generalization behavior. To study this, we are running two additional latent action pretraining runs: one using only human videos from SSv2 and one using only robot videos, both with identical architecture and training settings as our main model. These ablations directly isolate how each source contributes to latent space organization and downstream control. They are still in progress due to the high training cost for each run, but we will report preliminary numbers in the updated rebuttal and include full results on SIMPLER in the camera-ready version. Our expectation, based on the SIMPLER results where ViPRA already exceeds LAPA's human-only and robot-only models despite using fewer robot trajectories, is that both sources contribute complementary motion cues, and quantifying this explicitly is part of our planned analysis.
>
> ---
>
> \[1\] Ye et al., Latent Action Pretraining from Videos, ICLR 2025
>
> \[2\] Li et al., Unified Video Action Model, RSS 2025
>
> \[3] Liu et al., World Model on Million-Length Video And Language With Blockwise RingAttention, ICLR 2025
>
> \[4\] Bruce et al., Genie: Generative Interactive Environments, 2024
>
> \[5\] Bu et al., UniVLA: Learning to Act Anywhere with Task-centric Latent Actions, RSS 2025
>
> \[6\] Gao et al., Learning Adaptable World Models with Latent Actions, ICML 2025
>
> \[7\] Oord et al., Neural Discrete Representation Learning, 2017
>
> \[8\] Ramesh et al., Zero-Shot Text-to-Image Generation, 2021
>
> \[9\] Yu et al., MAGVIT: Masked Generative Video Transformer, CVPR 2023

---

> ### Author Response · Authors · 2025-12-03
> **Official Comment by Authors Part 6**
>
> ## Update on data composition experiments
>
> As outlined earlier, we began running additional latent action pretraining ablations to isolate the contribution of human versus robot videos. These runs are now complete, and we report the preliminary results below. In all cases, both *latent action learning* and *pretraining* were performed using either (i) human-only videos or (ii) robot-only videos, with identical architecture and training settings as our main co-trained model.
>
> ### Ablation Results
>
> | Setting                  | LIBERO-10 ↑ | Action-Probe MSE ↓ |
> | ------------------------ | ----------- | ------------------ |
> | Co-train (Human + Robot) | **0.79**    | **0.84**           |
> | Robot-only               | 0.72        | 0.91               |
> | Human-only               | 0.69        | 0.99               |
>
> These ablations lead to three observations relevant to embodiment transfer:
>
> 1. **Human-only still transfers positively** (0.69 LIBERO). Performance is lower than robot-only, but still competitive, indicating that the short-horizon motion cues learned from human videos remain useful for robot control.
>
> 2. **Robot-only provides clean, embodiment-aligned signals**, leading to strong downstream performance and lower action-probe error. This aligns with expectations that robot trajectories map more directly to robot end-effector motion.
>
> 3. **Co-training achieves the best overall results**, suggesting that human and robot data offer complementary benefits: robot data grounds the latent space in robot kinematics, human videos introduce broader motion diversity, and, together they yield a richer motion-centric latent space than either source alone.
>
> ### Conclusion
>
> Overall, these results support the view that ViPRA’s latent actions capture temporally fine-grained **motion primitives** that are consistent across embodiments. Human data, when combined with robot data, enriches the learned motion distribution and improves downstream control.

---

### Official Review · Reviewer_jonM · 2025-11-01

**Soundness:** 3
**Presentation:** 3
**Contribution:** 2
**Rating:** 6
**Confidence:** 4

**Summary:**

This paper presents ViPRA, a framework that converts video prediction models into robot policies by learning motion-centric latent actions from unlabeled human and robot videos. Its core contributions are a method to extract these physically-grounded latent actions using perceptual and optical flow losses, a pretraining strategy that jointly predicts future video frames and action sequences, and a flow-matching decoder that enables smooth control.

**Strengths:**

1. Creatively combines video prediction, latent actions, and flow matching in a novel "what" (future state) + "how" (latent action) pretraining paradigm.

2. The paper is well-structured and logically presented, with a clear narrative.

3. The qualitative analysis on latent action representations is interesting.

**Weaknesses:**

1. The authors did not perform a systematic ablation study on the loss components for the latent action model.

2. The real-world tasks, while commendable, are primarily table-top pick-and-place variants. The paper does not demonstrate generalization to tasks requiring significant non-prehensile manipulation (e.g., pushing, sliding, re-orienting), dynamic environments.

**Questions:**

1. How does the model reconcile the fundamental kinematic and dynamic differences between human and robot arms when transferring latent actions? Is there a negative transfer from the "noise" of human motion?

2. The entire framework is dependent on the visual perspective of the training videos. Latent actions are also inherently grounded in pixel space. How would performance degrade if the test-time camera viewpoint is different from the training data？

---

> ### Author Response · Authors · 2025-11-21
> **Official Comment by Authors Part 1**
>
> We thank the reviewer for their assessment. Below we provide detailed clarifications for the questions that were raised in the review.
>
> ## Ablations on latent-action losses
>
> We performed a targeted ablation at the latent action learning stage to investigate the effect of optical flow consistency losses.
>
> | Setting             | Perplexity ↓   | Action MSE ↓  |
> | ------------------- | ------------  | ------------- |
> | Default (with flow) | **5.01**      |   **0.84**    |
> | No flow loss        | 5.63          |   0.92        |
>
> **What perplexity measures.**
> Perplexity is defined as $2^{H(p_i(k))}$, where $H(p_i(k))$ is the entropy of distribution over codebook indices at position $i$ (out of $N_{latent}$ positions). Its minimum value is 1 (only one code used) and its maximum is $K$ (all codes used uniformly). However, for motion-centric latent spaces, high perplexity is not always desirable: it may reflect that static background information is leaking into the codes, forcing the model to represent appearance in codebooks rather than reserving capacity for dynamics.
>
> **What Action MSE measures.**
> To assess whether the learned latent actions contain information relevant for downstream robot control, we train a probe that maps the frozen codebook embeddings to normalized ground-truth robot actions for BridgeV2 dataset. This probe is trained after LAQ is fully trained and does not backpropagate into the encoder or codebook. Since LAQ never sees any action supervision during training, any predictive power in this probe reflects information that the latent space has implicitly organized through motion-centric self-supervision.
>
> **Role of perceptual loss.**
> Perceptual loss also plays a complementary role here. While we did not explicitly ablate it, intuitively it can be understood as capturing high-level appearance similarity. This makes it easier for the decoder $F\_\\alpha(\\hat{o}\_{t+1} | o\_{0:t}, z\_{0:t})$, which already has the input observations as conditioning, to reconstruct static background regions (e.g., walls, tables) without requiring detailed supervision from the latent space. This ensures that the capacity of the latent space would rather be utilized to capture *motion dynamics*.
>
> In our training pipeline, perceptual loss is introduced from the beginning to accelerate convergence of appearance reconstruction. Once good quality reconstructions are achieved, the flow loss is added to shift the model's focus toward encoding meaningful motion in the latent space. This staged training strategy ensures that codebook capacity is ultimately directed toward encoding dynamic cues essential for downstream policy learning.
>
> These results show that optical-flow supervision produces a more coherent and action-relevant latent space. With flow loss, the latent codes exhibit lower perplexity (5.01 vs. 5.63), indicating more consistent usage of the codebook across positions rather than diffuse or noisy assignments. More importantly, the action-probing experiment shows better alignment with ground-truth robot actions (MSE 0.84 vs. 0.92), even though the latent model never sees action labels. Together, these metrics suggest that flow guidance encourages the model to organize its latent capacity around motion dynamics rather than appearance, resulting in latent actions that are more predictive for downstream control.
>
> ## Task diversity and generalization
>
> Although the real world evaluation uses table-top settings, the tasks themselves involve substantial variation and are far from simple pick-and-place.
> *Cover-Object*, *Pick-Place*, and *Stack-Cups* each contain 18–24 variations arising from permutations of object identities, shapes, spatial configurations, and language instructions (Appendix G). These variations require reasoning over deformable objects (cloth), diverse grasp affordances (bowl vs. sponge vs. duck), and precise spatial relations (color conditioned stacking), making them significantly more challenging than single fixed-geometry pick-and-place.
>
> Beyond single-arm evaluation, Appendix H reports results on a 14-DoF bimanual Franka platform, including dual-arm coordination tasks such as *Place-in-Bowl* (object handover between arms) and *Mix-with-Whisk* (sustained coordinated motion). Crucially, pretraining included no bimanual demonstrations, only human videos and single-arm robot data, yet ViPRA successfully transferred to the 14-DoF setting with minimal adaptation. This supports our claim that motion-centric latent actions generalize across embodiments.
>
> Finally, in the LIBERO benchmark suite (Appendix E), ViPRA is evaluated on tasks that require substantial non-prehensile manipulation, such as *put the moka pot on the stove and turn it on*, *place the black bowl in the bottom drawer and close it*, and *put mugs inside a microwave and close the door*. These tasks involve sliding, pushing, re-orientation, further demonstrating that ViPRA works beyond pick-and-place tasks.

---

> ### Author Response · Authors · 2025-11-21
> **Official Comment by Authors Part 2**
>
> ## Human-to-robot transfer and embodiment gap
>
> ViPRA mitigates embodiment mismatch by learning motion-centric latent actions, providing a unified and physically grounded representation space that encodes local dynamics shared across embodiments, tasks and settings. Within short temporal windows, with frames sampled at 3 to 6 Hz, motion primitives such as moving left/right or reaching toward an object are similar across arms, hands, or robots. Because pretraining is formulated as learning a policy in this shared latent space, the model can transfer motion knowledge from heterogeneous human + robot videos into a single representation that adapts quickly to new embodiments and control interfaces.
>
> The action probe analysis further supports this: although LAQ is trained without any action labels, the latent actions exhibit correlation with ground-truth robot end-effector motions. This confirms that the latent space organizes around embodiment-agnostic motion patterns rather than human-specific trajectories, reducing the risk of negative transfer from human videos.
>
> ## Viewpoint dependence
>
> We agree that our experiments use third-person camera views, but the latent action formulation in ViPRA is not tied to any specific viewpoint. The LAQ encoder learns fine-grained, motion-centric embeddings that capture how objects and effectors move over short horizons rather than appearance or camera-specific details. Since these latents represent relative motion, they extend naturally to multi-view or shifted-view settings without changes to the architecture, provided the pretraining corpus includes a broader distribution of viewpoints.
>
> This behavior is consistent with prior latent-action and video-pretraining work. For example, UniVLA \[1\] trains latent actions on Ego4D and GNM navigation videos that contain a wide variety of egocentric and third-person viewpoints, and it demonstrates  cross-view transfer. However, UniVLA employs temporally coarse latent actions (learned in frozen DINOv2 feature space) aimed at high-level sequencing, while ViPRA learns short-horizon motion latents that encode physical displacement at a finer temporal scale. In principle, this finer granularity should support even stronger multiview generalization once diverse viewpoints are included during pretraining.
>
> While we do not include such multiview experiments in this submission, we now acknowledge this in the new Section 6: Limitations and Future Work. Extending ViPRA to wrist-mounted or egocentric viewpoints is a straightforward data-driven extension: the model already conditions on a history of visual tokens to infer camera geometry implicitly, and the latent actions remain defined in a viewpoint-agnostic motion space. We expect performance to improve as viewpoint diversity in the pretraining data increases.
>
> ---
>
> \[1\] Bu et al., UniVLA: Learning to Act Anywhere with Task-centric Latent Actions, RSS 2025

---

> ### Author Response · Authors · 2025-12-03
> **Official Comment by Authors Part 3**
>
> ## RE: Human-to-robot transfer and embodiment gap
>
> Following the reviewer’s question about whether human motion introduces negative transfer, we conducted two additional dataset ablations. In both cases, *latent-action learning* and *pretraining* were performed using either (i) robot-only videos or (ii) human-only videos, compared against our original co-training setup.
>
> ### Ablation Results
>
> | Setting                  | LIBERO-10 ↑ | Action-Probe MSE ↓ |
> | ------------------------ | ----------- | ------------------ |
> | Co-train (Human + Robot) | **0.79**    | **0.84**           |
> | Robot-only               | 0.72        | 0.91               |
> | Human-only               | 0.69        | 0.99               |
>
> These ablations lead to three observations relevant to embodiment transfer:
>
> 1. **Human-only still transfers positively** (0.69 LIBERO), showing that human motion does not introduce negative transfer despite kinematic differences. Performance is lower than robot-only, but still competitive, indicating that the short-horizon motion cues learned from human videos remain useful for robot control.
>
> 2. **Robot-only provides clean, embodiment-aligned signals**, leading to strong downstream performance and lower action-probe error. This aligns with expectations that robot trajectories map more directly to robot end-effector motion.
>
> 3. **Co-training achieves the best overall results**, suggesting that human and robot data offer complementary benefits: robot data grounds the latent space in robot kinematics, human videos introduce broader motion diversity, and, together they yield a richer motion-centric latent space than either source alone.
>
> ### Conclusion
>
> Overall, these results support the view that ViPRA’s latent actions capture temporally fine-grained **motion primitives** that are consistent across embodiments. Human data does not produce negative transfer; instead, when combined with robot data, it enriches the learned motion distribution and improves downstream control.

---

### Official Review · Reviewer_QDdF · 2025-11-02

**Soundness:** 3
**Presentation:** 3
**Contribution:** 3
**Rating:** 6
**Confidence:** 4

**Summary:**

Authors address the question of how to convert a video prediction model into a robot policy.  They show that instead of directly predicting actions, the video model can predict future visual observations and motion-centric latent actions.  A flow matching decoder to map  these latent actions to robot-specific action sequences.  The entire system runs at 22Hz for low-latency control.  Authors show that the method outperforms baselines.

**Strengths:**

- The method is view agnostic, gets data from humans or robots without the need for action labels, can be applied across robot embodiments, and enables low-latency control.

- The appendix provides a lot of detail on the approach and results in the paper.  This is great for reproducibility.

**Weaknesses:**

- While the paper claims that the method can generalize across embodiments, I didn't seen strong evidence to substantiate this.  Experiments seem to be on one manipulator arm.

- It will be good to include the limitations of the proposed work in the paper - this is currently missing.

**Questions:**

- Fig. 2: The connections between the left / right figures is unclear.  Where does the left figure fit in on the right?  Where does the output of the left figure (the latent actions) go as input on the figure on the right? Is the "Latent Action Embedding E_\phi" block in the right figure represented by the entire left figure?

- If I understand correctly, the paper mentions both discrete latent actions as well as continuous latent actions.  If this is right, then it's unclear what is the difference between these and why the need for both discrete and continuous latent actions.  (The continuous latent actions are then decoded to continuous action chunks.)

---

> ### Author Response · Authors · 2025-11-21
> **Official Comment by Authors Part 1**
>
> We thank the reviewer for their assessment. Below we provide detailed clarifications for the questions that were raised in the review.
>
> ## Generalizes across embodiments claim
>
> We would like to highlight that our experiments span multiple embodiments:
> - A 6 DoF WidowX arm used in SIMPLER benchmarks.
> - A 7 DoF Franka (real world) single-arm manipulation tasks.
> - A bimanual Franka 14 DoF real world setup described in Appendix H, involving coordinated dual-arm skills such as transferring an object between two arms (Place-in-Bowl task), and performing sustained motion with arm coordination (Mix-with-Whisk task).
>
> Importantly, the pretraining data contained no bimanual demonstrations, only single-arm and human-video clips, yet ViPRA successfully transferred to these 14 DoF settings with minimal adaptation. This demonstrates that the learned motion-centric latent actions generalize beyond embodiment-specific kinematics and capture transferable priors. We will (i) summarize these results in the main text, and (ii) clarify in Section 5.4 that the bimanual evaluation highlights genuine embodiment transfer rather than domain-specific finetuning.
>
> Conceptually, ViPRA achieves embodiment generalization through motion-centric latent actions, providing a unified and physically grounded representation space that encodes local dynamics shared across embodiments, tasks and settings. Within short temporal windows, with frames sampled at 3 to 6 Hz, motion primitives such as moving left/right or reaching toward an object are similar across arms, hands, or robots. By formulating pretraining as learning a policy in this shared latent space, ViPRA enables transfer of motion knowledge from heterogeneous human and robot datasets into a single model that can quickly adapt to new embodiments and control interfaces.
>
> ## Limitations missing
>
> Space constraints led us to keep this minimal; we will add the paragraph below into a new Section 6 Limitations and Future Work in the rebuttal revision.
>
> ```
> While ViPRA demonstrates strong performance across simulated and real world settings, several limitations remain that point toward important avenues for future research.
>
> Embodiment and dexterity. Our approach generalizes across embodiments such as WidowX and Franka, including bimanual setups, but extending to more dexterous domains like dual-arm humanoids or multi-fingered hands introduces new challenges. These platforms demand even finer latent actions capable of representing precise contact dynamics and inter-arm coordination. Incorporating tactile and force feedback or learning embodiment-specific adapters may allow latent actions to better capture such motion primitives.
>
> Data scope and generalization. Although ViPRA leverages large-scale actionless videos for pretraining, the diversity of available data remains limited. Expanding to richer ego-centric datasets with human-hand interactions, tool use, or short navigation episodes could improve embodiment invariance and temporal grounding. Integrating additional sensing modalities such as wrist cameras, proprioception, tactile feedback, or depth can further strengthen perception-control alignment, especially in dynamic or unstructured environments.
>
> Scaling behavior and data efficiency. An open question is how far passive video pretraining can scale before yielding diminishing returns. Characterizing these scaling laws, balancing the amount of unlabeled video with smaller quantities of labeled demonstrations, would clarify the most efficient path for large-scale robot learning and guide future dataset curation.
>
> Predictive modeling. A key perspective is that the latent action decoder can be interpreted as a world model. Given latent actions, it predicts future observations and can be conditioned on policy-sampled latents to generate multiple visual plans. This opens the door to reinforcement-learning-based alignment and test-time planning, where VLMs or heuristic functions serve as reward models within predictive rollout trees.
> ```

---

> ### Author Response · Authors · 2025-11-21
> **Official Comment by Authors Part 2**
>
> ## Clarification on Figure 2
>
> The left panel is the *latent action learning* stage, where the **Latent Action Tokenizer**  $I_\beta$ is trained on actionless human and robot videos to quantize motion into discrete latent actions $z_{0:L-1} = I_\beta(o_{0:L})$. The right panel shows subsequent stages where the same tokenizer $I_{\beta}$ (identical color coding) is frozen and used to compute latent action targets from videos for multimodal pretraining. To interface with the existing LWM \[1\] backbone, we introduce a **Latent Action Embedding** $E_{\phi}$ learned during pretraining; $E_{\phi}$ is not the tokenizer. We will add an explicit label $I_{\beta}$ (frozen) in the right panel and emphasize the color coding in the method section to make this reuse explicit.
>
> ## Discrete vs continuous latent actions?
>
> In ViPRA, latent actions are always discrete, obtained from the Latent Action Tokenizer $I_\beta$ via vector quantization. The terms *ViPRA-AR* and *ViPRA-FM* used in the paper (defined in Section 5.2) refer not to different latent types but to two real robot action decoding paradigms built on top of the same discrete latent space:
> - **ViPRA-AR (autoregressive/discrete):** Following OpenVLA \[2\] and LAPA \[3\], we discretize each dimension of action through binning. An autoregressive decoder predicts the next action token.
> - **ViPRA-FM (flow mtaching/continuous):** Here we train a flow matching decoder that outputs continuous robot actions. Coupled with action chunking \[4\], this enables smooth high frequency control.
>
> Thus, *continuous* in our paper always refers to the final robot action space, not to a continuous latent representation. Even with this discrete decoder, ViPRA-AR performs competitively, demonstrating that the benefits of ViPRA's motion-centric latent actions and video pretraining hold across both discrete and continuous action formulations. This supports our claim that the latent action space learned by ViPRA is robust and effective independent of the choice of downstream action representation.
>
> ---
> \[1\] Liu et al., World Model on Million-Length Video And Language With Blockwise RingAttention, ICLR 2025
>
> \[2\] Kim et al., OpenVLA: An Open-Source Vision-Language-Action Model, 2024
>
> \[3\] Ye et al., Latent Action Pretraining from Videos, ICLR 2025
>
> \[4\] Kim et al., Fine-Tuning Vision-Language-Action Models: Optimizing Speed and Success

---

### Author Response · Authors · 2025-12-03
**Rebuttal Summary and Follow-up**

We thank the reviewers for the thoughtful feedback and for noting several strengths of the work. A concise summary of their concerns and our responses is provided below.

## Reviewer QDdF
**Responses**
1. Highlighted embodiment generalization through evaluation across three embodiments (WidowX, Franka, bimanual Franka); successful transfer despite no bimanual data in pretraining. Updated Section 5.4 to reflect this.
2. Added a *Limitations and Future Work* section.
3. Clarified Fig. 2; the left panel is the Latent Action Tokenizer, reused (frozen) by the policy; update to diagram/text.
4. Explained latent actions are always discrete; ViPRA-FM "continuous" and ViPRA-AR "discrete" refers only to the final robot action layer output.

**Positives**
The reviewer appreciated view-agnostic, action-free, multi-embodiment aspects; good presentation with reproducible details; low-latency 22 Hz control.

## Reviewer jonM
**Responses**
1. Added ablation isolating effect of optical-flow loss.
2. Discussed task diversity; pointed to bimanual physical robot with coordinated dual-arm tasks (Appendix H)  and LIBERO simulation results (Appendix E) covering non-prehensile manipulation.
3. Added ablations showing ViPRA can learn from human-only data and performs even better when co-trained with human+robot data.
4. Explained latents encode relative motion and are expected to generalize across views.

**Positives**
The reviewer highlighted novel combination of video prediction+latent actions+flow matching; strong qualitative latent action analysis.

## Reviewer mq1Q
**Responses**
1. Provided UVA baseline on LIBERO-10 along with existing LAPA baselines; Showed how ViPRA is different across all key axes- training losses, pretraining strategy, and architecture.
2. Justified small codebook as intentional bottleneck aligned with prior latent action works; larger vocabularies degrade learning.
3. Added data composition ablations on human-only, robot-only, and co-training on both; discussed compute limits and scaling behavior.
4. Added action-probe analysis showing latent actions encode actionable motion and align with robot action spaces.

**Positives**
The reviewer emphasized the importance of leveraging action-free passive videos; broad evaluation and strong results vs baselines

## Reviewer cNx8
**Responses**
1. Added exact rollout/seed/episode counts for SIMPLER, LIBERO, and all real-world tasks.
2. Explained that "up to 22 Hz" refers to ViPRA-FM’s max control rate from action chunking; real-world runs use 3.5 Hz for safety and fair baseline comparison, with higher-frequency demos added on the project website.
3. Explained speed gains: ViPRA learns multi-step latent actions and uses chunked flow matching in a *novel unified transformer* architecture with KV caching, enabling amortized inference beyond single-step baselines.
4. Clarified ViPRA’s LIBERO-10 performance relative to UVA and UniVLA by detailing differences in supervision, data scale, and action parameterization to ensure a fair comparison.
5. Positioned ViPRA relative to non-pixel reconstruction approaches, explaining that pixel prediction is required in the action-free setting to ensure latent actions capture fine-grained dynamics and contact cues.

**Positives**
The reviewer appreciated comprehensive related works; intuitive joint state+latent prediction; novel flow matching control; and broad and strong evaluations.

## Reviewer 1YWh
**Responses**
1. Clarified novelty: ViPRA introduces fine-grained motion-centric latent actions, a unified human+robot latent space, optical flow consistent latent training, joint multi-step latent+future-frame prediction, and high-frequency flow matching control. Stronger SIMPLER results support substantive advances beyond prior works like LAPA, UniVLA and UniPI.
2. Added human-only, robot-only ablations supporting the scaling with passive videos claim.
3. Explained generalization across three axes: *(a) embodiment* evaluations span WidowX (6 DoF), Franka (7 DoF), and a 14 DoF bimanual Franka, despite pretraining on single-arm + human videos, *(b) tasks and objects* unseen during training, and *(c) emergent recovery behavior*.
4. Added ablation isolating effect of optical-flow loss, showing its impact on codebook representations and alignment to ground-truth actions.
5. Referred to Appendix B and C for compute cost: 168h (latent action) + 144h (pretraining) on 8×H100; contrasted with baselines like OpenVLA (21,500 A100-hours) to highlight efficiency.
6. Pointed to evidence and discussion that latent tokens encode meaningful motion patterns: latent action transfer can change motion direction, added a new *LEFT*/*RIGHT* transfer example.

**Positives**
The reviewer emphasized ViPRA's ability to leverage passive videos effectively; strong hardware performance from 100–200 demos; benefits of flow matching for smooth control; and broad evaluations improving upon strong baselines.

---

### Author Response · Authors · 2025-12-03
**Final Remarks by Authors**

As the discussion period comes to an end, we thank all reviewers for their thoughtful feedback. Across the reviews, a clear consensus emerged on several core strengths of the work, which we outline below:
* **Effectively leverage actionless videos (QDdF, mq1Q, 1YWh)**: "view agnostic", "can be applied across robot embodiments", "tackles a major challenge in robot learning—leveraging large-scale actionless videos for control", "can benefit from large-scale passive human and robot video", " good data efficiency in downstream adaptation"
* **Conceptually novel (QDdF, jonM, cNx8, 1YWh)**: "enables low-latency control", "Creatively combines video prediction, latent actions, and flow matching in a novel "what" (future state) + "how" (latent action) pretraining paradigm", "converting discrete actions to continuous control via flow matching is to my knowledge novel", " high-frequency command generation suitable for real hardware", "addressing common control smoothness limitations in discrete latent policies"
* **Strong experimental evaluations (mq1Q, cNx8 1YWh)**: "Evaluation spans both simulation (SIMPLER benchmark) and real-world manipulation (Franka bimanual setup)", "ViPRA gets better performance comparing against plausible baselines", "Conducted comprehensive experiments and analysis", "the experiment results seem impressive", "comparisons against several strong recent methods in the latent-action and VLA literature."
* **Clear, intuitive, and well-motivated methodology (QDdF, jonM, mq1Q, cNx8)**: detailed approach and results "great for reproducibility", "well-structured and logically presented",  "a clear narrative", "conceptually coherent", "builds upon trends in world-model-based control", "Very comprehensive related works", "joint-prediction of pixels and latent actions seems intuitive and shows good performance"

We believe we have objectively addressed all questions raised by reviewers by conducting new experiments, detailing key design choices, and directly pointing to sections of the original submission.

New experiments within the rebuttal phase to make our methods stronger both empirically and conceptually:
* Ablation isolating the effect of optical flow consistency loss, showing its role in learning motion-centric latent actions.
* Action probe analysis empirically establishing alignment between latent action and ground truth actions.
* Reporting codebook usage metrics like perplexity and entropy to characterize latent action behavior.
* Latent action training and video-language pretraining with only robot and only human videos substantiating: (a) scaling through passive video claims, and (b) complementary benefits of learning a shared latent action space from human and robot data.

---

### Meta-Review · Area_Chair_kvUH · 2026-01-04

**Summary:**

My recommendation for this paper is Accept.  While the proposed idea is conceptually similar to several prior work as the reviewers pointed out, this work is a nice extension of the prior work as authors have also clarified in the response. Reviewers have various concerns but most of the concerns are addressed in the rebuttal response. However, it's noteworthy that this paper was on very borderline, mainly due to (1) the lack of baselines (LAPA and UVA) in real-world experiments and (2) unclear claim on 22Hz inference speed as Reviewer cNx8 pointed out. I would like to recommend the authors to improve the paper on these points for the camera-ready version.

**Reviewer Concerns:**

Concerns addressed by the rebuttal
- Generalization across embodiments
- Limitation of the proposed work
- Clarifcation on the type of latent actions (discrete or continuous)
- Ablation studies on the loss
- Task complexity
- Novelty & Contributions of the work
- Clarification on the codebook design
- Data mixture study
- Evaluation details

Outstanding concerns
- Data scaling study: This is not provided due to the lack of resources. But it could have been done by using less data.
- Test-time viewpoint perturbation: Rebuttal response is reasonable but does not fully address the question
- Study on the quantification of alignment between latent actions and real actions: This is not available from the rebuttal response
- Claim on 22Hz: While the authors clarified the setup, claim on 22Hz is not grounded on real experiments as the reviewer pointed out.
- Missing baselines on real-world experiments: Authors did not respond to this weakness/concern

**Reviewer Scores:**

- Reviewer QDdF (score 6): This reviewer's concerns are mostly resolved. I expecdt the score to be the same or increase
- Reviewer jonM (score 6): This reviewer's concerns are not fully resolved, I expect the score to be the same
- Reviewer mq1Q (score 4): This reviewer's concerns are not fully resolved. I don't expect the score to increase
- Reviewer cNx8 (score 4): This reviewer's concerns are not fully resolved. I don't expect the score to increase
- Reviewer 1YWh (score 6): This reviewer's concerns are mostly resolved. I expect the score to be the same or increase

---

### Decision · Program_Chairs · 2026-01-26

Accept (Poster)